# New Recipe for Semi-supervised Community Detection: Clique Annealing under Crystallization Kinetics

## Abstract

Semi-supervised community detection methods are widely used for identifying specific communities due to the label scarcity. Existing semi-supervised community detection methods typically involve two learning stages *i.e.*, learning in both initial identification and subsequent adjustment, which often starts from an unreasonable community core candidate. Moreover, these methods encounter scalability issues because they depend on reinforcement learning and generative adversarial networks, leading to higher computational costs and restricting the selection of candidates. To address these limitations, we draw a parallel between crystallization kinetics and community detection to integrate the spontaneity of the annealing process into community detection. Specifically, we liken community detection to identifying a crystal subgrain (core) that expands into a complete grain (community) through a process similar to annealing. Based on this finding, we propose CLique ANNealing (CLANN), which applies kinetics concepts to community detection by integrating these principles into the optimization process to strengthen the consistency of the community core. Subsequently, a learning-free Transitive Annealer was employed to refine the first-stage candidates by merging neighboring cliques and repositioning the community core, enabling a spontaneous growth process that enhances scalability. Extensive experiments on diverse community detection datasets demonstrate that CLANN outperforms state-of-the-art methods across multiple real-world datasets, showcasing its exceptional efficacy and efficiency in community detection.

## 1 Introduction

Community detection aims to distinguish node groups with closer inner connections. (defined by concrete contexts) (Jeong et al., 2021; Li et al., 2019; Zhang et al., 2018; Abbe, 2023) Unsupervised methods eliminate the need for costly data labeling and are widely utilized due to the label scarcity in community detection (Holland et al., 1983; Amini et al., 2013; de Lange et al., 2014). While these methods have demonstrated strong performance and practicality, they often struggle to accurately identify specific communities with distinct semantic meanings. For instance, in a social network with 100 user communities, only 10 of which are fraudulent, unsupervised methods might identify all 100 communities but typically struggle to differentiate which ones are fraudulent. This is mainly because they are typically designed based on general structural information rather than the specific inherent features of the targeted communities.

To improve the detection of communities with semantic meaning, some semi-supervised methods have been introduced that primarily utilize a two-stage approach (Wu et al., 2022; Bakshi et al., 2018). They identify potential community centers first, then expand these centers into final communities. Although semi-supervised methods are intuitive, they still have the following limitations:

**Community Core Inconsistency**: Prevalent growth-based methods scan all vertices (or their k-ego networks) and calculate embedding space distances to the labeled communities to select optimal community cores. However, a single node feature is insufficient to represent structural information. Moreover, the k-hop ego network frequently includes vertices positioned outside of any community. Instead, as shown in App. A, a clique (where all nodes are connected and cannot be expanded by adding

another vertex) more accurately and essentially reflects cohesiveness in a given structure (Gupta & Singh, 2023; Maity & Rath, 2014; Mimaroglu & Yagci, 2012; Jia et al., 2019). Consequently, using cliques as starting points is more effective than traversing all nodes to identify community centers (Shen et al., 2009; Lu et al., 2010; Svendsen et al., 2015).

**Inferior Growth Scalability**: Besides, existing growth models often employ Reinforcement Learning (RL) modules to expand first-stage candidates using predefined reward functions (Wu et al., 2022; Zhang et al., 2020). However, these reward functions are typically disconnected from the design of the first stage, resulting in inefficient use of the initial information. Moreover, when Generative Adversarial Networks (GANs) are introduced to generate more realistic reward signals, they exacerbate scalability issues (Zhang et al., 2017), limiting the number of community core candidates and often leading to suboptimal solutions.

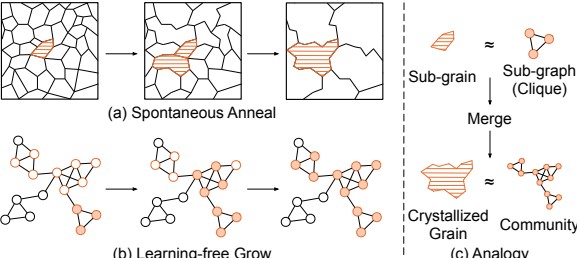

Figure 1: (a) Spontaneous annealing process where the subgrains grow into the crystallized grain by merging with other grains. (b) CLANN Schematic diagram. The initial clique spontaneous grows into a community by merging other cliques. (c) Analogy between community detection and annealing process.

To address the above challenge, we utilize the annealing process in crystallization kinetics to effectively simulate the seed-growth mechanism, allowing subgrains to merge with others and grow into fully crystallized grains, as illustrated in Fig. 1 (a). The annealing seeds are consistently subgrains, not just any random region. This aligns with the concept that a community core is not a random node or its K-ego network. Given that the entire crystallization process is spontaneous and follows physical laws, we further proposed CLique ANNealing (CLANN), which consists of two main components: the Nucleus Proposer and the Transitive Annealer. As illustrated in Fig. 1 (b), we first identify a clique as community core and gradually merges neighbor cliques to form the final community. The analogy between crystallization and community formation is illustrated in Fig. 1 (c).

Specifically, we first only optimize a single graph encoder to inherently analogize community formation by integrating four crystallization principles (Stability, Cohesion, Growth, and Status) into the optimization to mitigate **Community Core Inconsistency**. Instead of evaluating all individual nodes, our Nucleus Proposer selects the most prospective cliques, thus speeding up the selection process. To address the **Inferior Growth Scalability**, we further propose a learning-free Transitive Annealer that directly leverages the module trained in the Nucleus Proposer to guide the annealing of candidates into communities. This method circumvents the convergence challenges and high computational costs typically associated with reinforcement learning or GANs. Given the clique's high homophily scores (as shown in App. A), we choose the clique as the core motif in this work; other motifs may also be effective depending on the dataset. In summary, our contributions can be summarized as follows:

- We introduce a novel model, CLANN, that leverages the annealing process to detect communities providing a new physics-grounded perspective for community detection by studying the subgrain growth process.

- We introduce two key components in CLANN: Nucleus Proposer, which uses crystallization principles and cliques to learn graph representations and identify community cores, and Transitive Annealer, which ensures spontaneous growth guided by the Nucleus Proposer.

- Empirical evaluations on real-world datasets consistently show that CLANN outperforms state-of-the-art methods by a significant margin, demonstrating its effectiveness and efficiency across diverse network analysis scenarios.

## 2 RELATED WORK

Given the limited availability of labeled data, we concentrate on unsupervised and semi-supervised approaches for overlapping community detection, while also providing an introduction to clique-based methods.

## 2.1 Overlapping Community Detection with Un/Semi-supervised Methods

**Unsupervised Methods**. Unsupervised methods are particularly valuable for exploratory data analysis, especially in scenarios where no supervision information is accessible. NOCD (Shchur & Günnemann, 2019) uses a generative model for inferring community affiliations. Community-GAN (Jia et al., 2019) applies GANs to generate motifs and optimize vertex embeddings, representing membership strength. ACNE (Chen et al., 2021) employs a perception-based walking strategy and a discriminator to jointly map node and community embeddings. DFuzzy (Bhatia & Rani, 2018) uses a stacked sparse autoencoder to evolve overlapping and disjoint communities via modularity. BigClam (Yang & Leskovec, 2013) identifies densely connected overlaps to improve accuracy and scalability. ComE (Cavallari et al., 2017) enhances detection through a synergistic loop between community and node embeddings. vGraph (Sun et al., 2019) utilizes a mixture model to represent nodes as combinations of communities.

**Semi-supervised Methods**. In contrast to unsupervised learning methods, which impose strict limitations on pinpointing specific types of communities, semi-supervised methods can effectively utilize labels from previous community members making them more practical in identifying specific community types. BigClam-A (Bakshi et al., 2018) stands for BigClam-Assisted with graphs modified by adding extra edges between nodes in the same community. SEAL (Zhang et al., 2020) generates seed-aware communities using a Graph Pointer Network with incremental updates (iGPN). DGL-FRM (Mehta et al., 2019) captures community membership strength and sparse node. LGVG (Sarkar et al., 2020) is designed to learn multi-layered and gamma-distributed embeddings, allowing it to detect communities at both fine-grained and coarse-grained levels. Bespoke (Bakshi et al., 2018) leverages community membership information and node metadata to identify unique patterns in communities beyond traditional structures. CLARE (Wu et al., 2022) builds a locator to find the community seeds and uses a rewriter to modify the candidates. Although the aforementioned semi-supervised methods perform well, many of them focus heavily on architectural design while underutilizing the inherent graph structures. On the contrary, CLANN can fully exploit the insights of community positioning and formation mechanisms from inherent motifs.

## 2.2 Clique-based Methods

A clique is a complete subgraph, naturally capturing densely connected subgraphs. Clique-based methods are classified into K-clique-based and maximum-clique-based. K-clique methods (e.g., CPM (Palla et al., 2005), SCP (Kumpula et al., 2008), ECPM (Maity & Rath, 2014), WCPM (Zhang et al., 2017), LOC (Ma & Fan, 2019)) find and merge adjacent K-cliques into communities, while maximum-clique methods (e.g., EA/G (Zhang et al., 2005), MaxCliqueDyn (Konc & Janezic, 2007), GVG-Mine (Lee et al., 2012), ACENV (Cheng et al., 2018), PECO (Svendsen et al., 2015)) select cliques with the largest number of nodes as initial communities. Though these approaches utilize substructures, they often struggle to represent lower-dimensional embeddings while preserving structural complexity (Fan et al., 2020; Bo et al., 2020; Luo et al., 2020; Cheng et al., 2021). In contrast, CLANN generates insightful embeddings while maintaining formation mechanisms.

## 3 Problem Definition and Pipeline

**Problem Definition**: Give a graph $\mathcal{G} = (\mathcal{V}, \mathcal{E}, \mathcal{X})$, where the $\mathcal{V}$ represent the node set, $\mathcal{E}$ represents the edge set and $\mathcal{X}$ represents the node feature. The expert-labeled training communities are denoted as $C = \{C_1, ..., C_N\}$ where $N$ is the number of expert-labeled communities. The objective in semi-supervised community detection, is formulated as given a small set of expert-labeled communities $C^{train} = \{C_1, ..., C_m\}$ as training data, where $m$ ($m \ll N$), to predict $N$-$m$ communities $C^{pred}$ from $\mathcal{G}$. The predicted communities should be consistent with all rest labeled communities (test communities), $C^{test} = C \setminus C^{train}$.

**Pipeline**: As shown in Fig. 2, Nucleus Proposer first identifies cliques with embeddings similar to the training community embeddings as potential core candidates (represented by dashed circles). To get the final integral structure, the Transitive Annealer will expand candidates into communities. Considering community core inherently contains less information than an integral community, Nucleus Propose might inevitably locate the sub-optimal start point. The Nucleus Transition module is thus proposed to dynamically relocate the initial candidates to other better cliques.

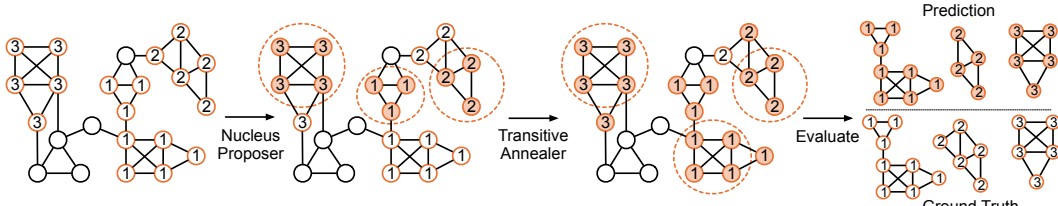

Figure 2: The pipeline of proposed model. Nodes with orange borders and identical labels belong to the same community. Dashed circles represent the currently predicted centers, while nodes filled in orange are those predicted to be within communities. It is worth mentioning overlapping communities are not shown for clarity, CLANN can also accommodate overlapping communities because core growth is driven solely by energy requirements, operating independently without considering previous community assignments.

## 4 NUCLEUS PROPOSER

The Nucleus Proposer aims to integrate the dynamics of community formation into an advanced graph encoder with crystallization kinetics. After training, it selects prospective cliques as community centers for further growth.

### 4.1 LINKING CRYSTALLIZATION TO COMMUNITY DETECTION

Many physics methods (Pang & Li, 2013; Greydanus et al., 2019; Cranmer et al., 2020) minimize a global energy function to penalize node assignments that do not correspond to natural communities. Since crystallization kinetics inherently reflect growth mechanisms where a community core can spontaneously expand into a full community without the need for learning, we apply these principles to simulate community growth and develop a more effective graph encoder, rather than relying on previous energy function. These principles can be clearly illustrated through four specific characteristics of communities, see more details in App. B.

**Community Stability**. Community stability correlates with size and outliers. Because members heighten interaction complexity and increase the potential for conflict. Outliers who deviate significantly from the norm can further disrupt stable dynamics. This is analogous to larger crystals with defects having higher stored energy, depicted in Fig. 3(a).

**Higher Cohesion**. Higher member similarity promises better community cohesion. Similar traits among members lead to harmonious interactions and stronger unity, akin to subgrains with the same crystallographic orientations merging in crystallization, as illustrated in Fig. 3(b).

**Spontaneous Growth**. Community growth consumes resources, as expanding a community involves resource expenditure to integrate infrastructure and complexity management. In crystallization, subgrains must overcome an Interface Energy Barrier to merge, with the total energy surpassing that of the larger, merged grain, as shown in Fig. 3(c).

**Equilibrium Status**. Larger and outlier-rich communities risk overgrowth and instability. Conversely, smaller and more cohesive communities have great potential for further growth (undergrown). This mirrors how a crystal's size and defect levels decide its status in Fig. 3(d).

### 4.2 IMPLEMENTATION OF CRYSTALLIZATION KINETICS

To implement the crystallization kinetics, we map each subgraph $g$ into a $d$-dimensional embedding $h(g)$ and ensure these principles can be reflected appropriately in the embedding space. Specifically, we construct positive and negative pairs with the subgraphs listed in Tab. 1 to optimize different loss functions. Positive pairs conform to these principles, while negative pairs violate them. The value of $m$ is given in App. H. We devise four novel loss functions to imitate the dynamics of community formation with crystallization kinetics.

Table 1: Sample notations.

| Notation | Definition |
| --- | --- |
| $a_1, a_2, a_3$ | Labeled community |
| $b, c$ | 1st/2nd largest clique in $a_1$ |
| $\bar{\circ}$ | Replace m% nodes in $\circ$ |
| $\dot{\circ}$ | Remove m% nodes from $\circ$ |
| $\breve{\circ}$ | one-hop neighbors of $\circ$ |

Figure 3: The connection between crystallization to community detection and the associated loss functions. We distill these principles into four core requisites: energy, consistency, interface, and integrity. S.E. and I.E., stand for stored and interface energy, respectively.

**Energy-Based Loss**. To capture the relationship between community stability and energy, we use the following loss function, which leverages the positive correlation between subgraph size and stored energy:

$$loss_{Size}^E = \sum_{(i,j)}^{Pos-S} \max\{0, ||i|| - ||j||\} + \sum_{(i,j)}^{Neg-S} \max\{0, \alpha - (||i|| - ||j||)\}, \quad (1)$$

where $\alpha$ is the loss margin, and $|| \cdot ||$ denotes the norm of the subgraph embedding $h$, which we utilize as an indicator of the graph's stored energy. We further define the positive and negative pair $Pos\text{-}S = \{(b, a_1), (c, a_1), (\bar{a_1}, a_1)\}$, $Neg\text{-}S = \{(a_1 + a_2, a_1), (a_2 + a_3, a_2), (a_3 + a_1, a_3)\}$, where '+' indicates combination. For each pair in $Pos\text{-}S$, the stored energy of the first subgraph is less than the second's, reflecting smaller subgraph sizes. Conversely, in $Neg\text{-}S$, the first subgraph, representing a merged subgraph, exhibits greater stored energy than the second subgraph due to its bigger size.

As the high defect concentration typically reflect the high stored energy, we further define the second energy-based loss function to describe the relationship between energy and misalignment:

$$loss_{Defc}^E = \sum_{(i,j)}^{Pos-D} \max\{0, ||i|| - ||j||\}, \quad (2)$$

where $Pos\text{-}D = \{(a_1, \bar{a_1}), (b, \bar{b}), (c, \bar{c})\}$. For each pair in $Pos\text{-}D$, the latter is distorted from the first subgraph (inside nodes are replaced with outside connected nodes). The second subgraph's stored energy should thus be larger than the first one.

**Consistency-Based Loss**. To ensure consistency, crystal grains with matching crystallographic orientation are expected to merge more easily into a unified grain. Similarly, major cliques within a community should follow the consistency requirement. Therefore, we require the aggregate of these clique embeddings $q_i$ to closely approximate the community embedding:

$$loss^C = d(h(a_1), \sum_i^{\lambda_{clq}} h(q_i)), \quad (3)$$

where $d(.,.)$ is a distance function defined in App. D, $\lambda_{clq}$ represents how many biggest cliques in $a_1$ are used to represent $a_1$. $\lambda_{clq} \leq |Q_{a_1}|$, where $Q_{a_1}$ is the set of all cliques in $a_1$. For instance, if we set $\lambda_{clq}$ to 2, the loss $loss^C$ will be formulated as $d(h(a_1), h(b) + h(c))$.

**Interface-Based Loss**. The energy barrier determines whether a subgrain can continue to grow. In community detection, we use a similar concept to decide if subgraphs can expand into a larger community. Specifically, the total stored energy of the subgraphs, combined with the interface energy, must exceed the stored energy of the resulting larger community (positive pairs). Conversely, when insufficient energy is available, a well-structured community cannot expand further (negative pairs). The interface-based loss function can thus be formed as:

$$loss^I = \sum_{(i,j)}^{Pos-I} \max\{0, ||i|| - ||j|| - \sum_{v \in \breve{j}} \text{Intf-E}(v)\} + \sum_{(i,j)}^{Neg-I} \max\{0, ||i|| - ||j|| + \sum_{v \in \breve{i}} \text{Intf-E}(v)\}, \quad (4)$$

where $Pos\text{-}I = \{(a_1, b), (a_1, c)\}$, $Neg\text{-}I = \{(a_1, a_1 + \breve{a_1})\}$, $\breve{*}$ stands for the one-hop neighbors. Intf-E$(\cdot)$ represents the interface energy between the neighbor node and the corresponding subgraph,

and is defined as follows:

$$\text{Intf-E}(v) = \text{Softplus}(W_I * f_v + b_I), \tag{5}$$

where $W_I$ and $b_I$ represent the weights and bias respectively, $f_v = [f_v^a || f_{v,g}^e]$, $f_v^a = [f_v^o, deg(v), max(DN(v)), min(DN(v)), avg(DN(v)), std(DN(v))]$ is the standard augment feature of node $v$, $f_v^o$ is the raw features of node $v$ with default value of 1, following the previous studies (Wu et al., 2022; Cai & Wang, 2018; Zhang et al., 2020). $deg(v)$ is the degree of node $v$ and $DN(v)$ represents the degree set of the neighbor nodes of $v$.

For the corresponding subgraph $g$, the external feature $f_{v,g}^e$ is represented as $[l\text{-}num, graph\text{-}size, mis\text{-}num]$, where $l\text{-}num$ is the edge number between node $v$ and $g$. $graph\text{-}size$ is the node number of $g$. It is worth noting that an ideal community is distinguished by high exclusivity and strong internal connections, forming a clique. Accordingly, $mis\text{-}num$ is defined as the number of nodes that are not part of a clique.

**Integrity-Based Loss**. The size and defect concentration play a crucial role in determining its growth status (overgrown, undergrown, or in equilibrium), which are essential for understanding how communities evolve and merge. To effectively integrate these factors, we propose an integrity-based loss function. The norm of the subgraph embedding represents its stored energy, while the normalized vector signifies defect concentration. Each subgraph is assigned a triplet integrity score, with [1,0,0], [0,1,0], and [0,0,1] representing the undergrown, equilibrium, and overgrown states, respectively. For subgraph $g$, its integrity score $\hat{y}_g$ is defined as the following:

$$p_g^k = \sigma(W_p^k * [h(g) || \bar{h}(g)] + b_p^k), k \in \{1, 2, 3\},$$
$$\hat{y}_g = [\hat{y}_g^1, \hat{y}_g^2, \hat{y}_g^3] = \text{Softmax}(p_g^1, p_g^2, p_g^3), \tag{6}$$

where $\sigma$ stands for activation function, $||$ stands for concatenation, $h(*)$ and $\bar{h}(*)$ are the graph embedding and normalized embedding for the given subgraph. $W_p^k$ and $b_p^k$ are the weights and bias of the corresponding network. The integrity-based loss function can thus be formed as:

$$loss^G = -\frac{1}{3} \sum_S^{\{S_1, S_2, S_3\}} \sum_{g \in S} \sum_{k=1}^3 (y_g^k \log(\hat{y}_g^k) + (1 - y_g^k)\log(1 - \hat{y}_g^k)), \tag{7}$$

where $y_g^k$ is the $k$-th dimension of $g$'s label, $\hat{y}_g^k$ is the corresponding prediction of $y_g^k$. $S_1 = \{b, c, \acute{a}_1\}$ stands for those subgraphs can still growth. $S_2 = \{a_1, a_2, a_3\}$ stands for stable subgraphs, and $S_3 = \{a_1 + \breve{a}_1, a_1 + a_2, a_2 + a_3, a_3 + a_1\}$ stands for overgrown subgraphs.

## 4.3 GRAPH ENCODER

The original node representation $f_v^a$ of node $v$ is transformed through a fully-connected layer into $z^0(v)$. Subsequently, the encoder disseminates and amalgamates the information through a $K$-layer GCN:

$$z^k(v) = \text{GNN}(z^{k-1}(v)), \ z^0(v) = \sigma(W^f f_v^a + b^f),$$
$$z(v) = \sigma(W^a * ||_{k=0}^K z^k(v) + b^a), \tag{8}$$

where $z^k(v)$ stands for the embedding of node $v$ after $k$ GNN layers, $z(v)$ is the final node embedding of node $v$ by concatenating all previous layer embeddings and transforming it with a linear layer. We then use sum-pooling $z(g)$ to represent the embedding for the given subgraph $g$.

In the preferential attachment model, new nodes tend to connect to high-degree nodes, and smaller cliques cluster around larger ones, forming a hierarchical structure. Hyperbolic geometry is effective for preserving tree-like structures between cliques and communities, making it effective for community detection (Gerald et al., 2023; Chami et al., 2020; Cao et al., 2022). We thus transform Euclidean graph embedding $z(g)$ to hyperbolic embedding $h(g)$, see App. D for more details. The overall loss function $\mathscr{L}$ is formulated as follows:

$$\mathscr{L} = \gamma^E loss^E + \gamma^C loss^C + \gamma^I loss^I \tag{9}$$

where $\gamma^{\{E,C,I\}}$ are the coefficients that regulate the balance between the contributions of different loss functions. $loss^E$ is the summation of $loss_{Size}^E$ and $loss_{Defc}^E$. After the initial training, we employ an independent fully connected layer to minimize $loss^G$ for status checking. Finally, cliques with

the shortest embedding distance to the training communities are chosen as the first-stage candidates. The algorithm of the Nucleus Proposer is provided in Algo. 1. (For large graphs, finding all cliques is extremely time-consuming. Therefore, we implement an preliminary selection mechanism prior Nucleus Propose, see App. F for more details.)

## 5  TRANSITIVE ANNEALER

To develop this core candidate selected by Nucleus Proposer into the final communities, we introduce a learning-free propagation method called Transitive Annelar. The pipeline of the Transitive Annealer and complexity analysis can be found in Algo. 2 and App. C. The meanings of the notations can be found in Tab. 2. In each growth iteration, we tackle three key points to ensure the candidates develop into reasonable structures.

### 5.1  POTENTIAL FOR FURTHER GROWTH OF THE CURRENT STATE

The integrity score and interface energy barrier are crucial for ensuring a community remains stable and suitable for further expansion. Therefore, we use integrity checks and interface energy assessments to evaluate community stability and determine the conditions for merging new subgraphs.

**Integrity Check**. Annealer expands candidates to communities iteratively. For the $m$-th iteration, we need to check the current state $S_{m-1}$'s integrity scores by Eq. 6. If $\hat{y}^1_{S_{m-1}} < \max(\hat{y}^{1,2,3}_{S_{m-1}})$, we deem the $S_{m-1}$ has already reached the stable or overgrown state, we thus stop the growth.

Table 2: Element notations. N, S, E, I stand for node, subgraph, energy, and integrity score.

| Notation | Type/Definition |
|---|---|
| $V^e$ | [N] $S_{m-1}$'s inner boundary nodes |
| $C^e_i$ | [S] Merge all cliques contain $v^e_i$ as $C^e_i$ |
| $S_{m-1/m}$ | [S] Current / next step state |
| $S^c_i$ | [S] $C^e_i + S_{m-1}$, Candidate state |
| Intf-E($*$) | [E] Interface energy btw $*$ and $S_{m-1}$ |
| $\|*\|$ | [E] Stored energy of a subgraph |
| $\hat{y}^j_*$ | [I] $j$-th integrity scores of a subgraph |

**Interface Energy Check**. As mentioned in Sec. 4.2, to surpass the interface barrier, subgraphs need to consume extra energy. For an extendable node $v^e_i \in V^e$, we merge all cliques containing $v^e_i$ as $C^e_i$, the corresponding expanded candidate $S^c_i$ is constructed by combining $C^e_i$ with $S_{m-1}$. Due to the interface energy barrier, the stored energy summation of the current state $S_{m-1}$, merged clique $C^e_i$, and their interface energy Intf-E($v^e_i$) (defined in Eq. 5) should be larger than the stored energy of the expanded candidate. We therefore require the following interface energy constraint:

$$\|C^e_i\| + \|S_{m-1}\| + |C^e_i| * \text{Intf-E}(v^e_i) \geq \|S^c_i\|, \tag{10}$$

where $|C^e_i|$ is the number of nodes inside $C^e_i$, but outside $S_{m-1}$.

### 5.2  SELECTION OF CLIQUE FOR MERGING

To develop a more cohesive community with fewer outliers, we select the candidate with the highest stored energy as the next state $S_m$. The annealing process continues until it either reaches the overgrown state or the maximum step count is reached:

$$\|S^c_k\| = \arg\max_{i \in |V^e|}(\|S^c_i\|); \quad S_m = S^c_k. \tag{11}$$

However, merely selecting the merged clique with the highest stored energy may lead to a local optimum. To address this issue, we employ the simulated annealing algorithm, which accepts suboptimal solutions with certain probabilities. Specifically, the weighting probabilities are defined based on the energy differences between the potential expanded states and the initial states:

$$\{P_1, \ldots, P_{|V^e|}\} = \text{Softmax}(D_1, \ldots, D_{|V^e|}),$$
$$D_i = \|S^c_i\| - \|S_{m-1}\|. \tag{12}$$

With a pre-defined temperature probability function $P_{temp}(|S^c_i|)$, where $|S^c_i|$ is the node number of $S^c_i$. The candidate is selected if its weight score exceeds $P_{temp}(|S^c_i|)$ (the function will be illustrated in App. H). Accordingly, the updated state $S_m$ is determined as follows:

$$\hat{S}^c_i = \{S^c_i | P_i > P_{temp}(|S^c_i|)\}, \ (i \in \{1, \ldots, |V^e|\}),$$
$$S_m = \bigcup\{S^c_i | \mathbb{1}(C^e_i, S_{m-1}, S^c_i) = 1\}, \ S^c_i \in \hat{S}^c_i, \tag{13}$$

where the indicator function $\mathbb{1}(C^e_i, S_{m-1}, S^c_i) = 1$, if the requirement in Eq.10 is satisfied.

## 5.3 SHIFTING OF COMMUNITY CORE

The Nucleus Proposer utilizes cliques for matching, potentially resulting in sub-optimal community center selections. To dynamically identify prospective cliques during growth, we determine if a merged clique $C_i^e$ could serve as a superior community core by calculating its integrity scores using Eq. 6. If the integrity score $\hat{y}_{C_i^e}^2$ of $C_i^e$ surpasses that of the previous state $\hat{y}_{S_{m-1}}^2$ and all other extendable nodes, the center will be shifted to $C_i^e$ and a new annealing cycle will be initiated, as illustrated in Fig. 2, where the center of community 1 has been adjusted.

## 6 EXPERIMENT AND ANALYSIS

**Dataset**. Our datasets include **A**mazon (Product), **D**BLP (Citation), and **L**iveJournal (Social Network), each containing a graph and 5,000 labeled communities. We also compare with classic unsupervised methods in Table 13 and 15, and test model under Non-Clique (Bipartite) setting in Table 16. We used two experimental settings:

Setting 1 (from CLARE): We strictly replicated from Wu et al. (2022); Zhang et al. (2020) to ensure a fair comparison. In this setting, communities above the 90-th percentile in size were excluded, and 1,000 communities were then sampled. Additionally, 5,000 edges were added between two graphs from different datasets to create a hybrid graph (e.g., A/D). The A/D task was to identify communities only from A (No D community should be found) within the hybrid graph.

Setting 2: Concerning about the ability of finding communities under different sizes, in this setting, no community exclusions, link insertions, or hybrid networks. This setting can better study the impact of large community sizes. We sorted communities by size and conducted 15 experiments to evaluate the model's performance across different community sizes. The split setting: training (9%), validation (1%), and testing sets (90%). Detailed information about the datasets can be found in App. E.

**Baselines & Metrics**. We compare CLANN with the following models: BigClam (Unsupervised) Yang & Leskovec (2013) and its assisted version BigClam-A, ComE (Unsupervised) Cavallari et al. (2017), CommunityGAN Jia et al. (2019), vGraph (Unsupervised) Sun et al. (2019), Bespoke Bakshi et al. (2018), SEAL Zhang et al. (2020), and CLARE Wu et al. (2022). NP stands for Nucleus Proposer (directly using clique candidates as predictions). Top 4 models are selected for adaptability analysis. We follow the evaluation metrics (bi-matching F1, Jaccard, and ONMI) in Bakshi et al. (2018); Chakraborty et al. (2017); Jia et al. (2019), each metric's definition is provided in the App. G.

### 6.1 OVERALL PERFORMANCE

We compared CLANN with different baseline models in Table 3. CLANN significantly outperforms other models in various metrics. On single datasets, CLANN improves the F1 score by an average of $14.54\%$ over the top SOTA models; for hybrid datasets, the increase averages $46.74\%$, with scores nearly doubling those of the closest competitors in some cases. Probabilistic methods struggle on hybrid datasets as they rely on statistical distributions that align well within single datasets but fail to capture the nuances between two combined datasets.

Bespoke and SEAL rely on the initial core's quality, while CLARE lacks accuracy as many community cores don't fit the K-hop ego network structure. Additionally, most nodes lie

Table 3: F1 Scores (Jacc., OMNI in Table 9). A/D: find A's communities in A+D. Bold/Underline: 1st/2nd best scores. N/A: not converge in 2 days. We conduct 5 experiments for NP and CLANN, the average std is less than 0.0323.

| Model | A | D | L | A/D | D/A | D/L | L/D |
|---|---|---|---|---|---|---|---|
| BigClam | .6885 | .3217 | .3917 | .1759 | .2363 | .0909 | .2183 |
| BigClam-A | .6562 | .3242 | .3934 | .1745 | .2346 | .0859 | .2139 |
| ComE | .6569 | N/A | N/A | N/A | N/A | N/A | N/A |
| Com-GAN | .6701 | .3541 | .4067 | .0204 | .0764 | .0251 | .0142 |
| vGraph | .6895 | .1134 | .0429 | .0769 | .1002 | .0131 | .0206 |
| Bespoke | .5193 | .2956 | .1706 | .0641 | .2464 | .0817 | .1893 |
| SEAL | .7252 | .2914 | .4638 | .2733 | .1317 | .1906 | .2291 |
| CLARE | .7730 | .3835 | .4950 | .3988 | .2901 | .2480 | .2894 |
| NP | .7809 | .3979 | .3655 | .4586 | .3850 | .3334 | .2435 |
| CLANN | **.9055** | **.4701** | **.5144** | **.6578** | **.4355** | **.3373** | **.3932** |

outside labeled communities, adding noise to later stages. However, as Fig. 7 shows, almost all communities comprise internal cliques, enhancing the effectiveness of our Nucleus Proposer. Furthermore, most methods limit candidate numbers due to computational constraints, leading to sub-optimal centers. The Transitive Annealer, with its lower computational costs, can handle more candidates, reducing the likelihood of forming sub-optimal communities in the second stage.

## 6.2 ABLATION STUDY

**Effectiveness of Crystallization Kinetics**. We analyze the contribution of various loss functions in crystallization kinetics and the results are shown in Table 4. **Engy.** acts as a baseline, focusing on the graph's energy but neglecting interactions with neighboring nodes and inner component relationships. **Intf.** considers interactions between the graph and neighboring nodes, which is crucial for community integration and exclusion. **Cons.** aims to ensure embedding robustness by requiring the sum of the embeddings of a community's two largest cliques close to the community embedding. **Intg.** evaluates community status, although achieving minimal performance

Table 4: F1 scores (Jaccard, OMNI scores are in Tab. 10) of different schemes. **Engy.**, **Intf.**, **Cons.**, and **Intg.** stand for energy, interface, consistency, and integrity losses. (+: add new scheme). **NP**: Nucleus Proposer with hyperbolic geometry. **+Infc**: Filter out candidates by interface energy. **+SA**: Simulated Anneal. **+C-E**: Clique-wise operation. **+TA**: Transitive Annealer.

|  | Engy. | +Intf. | +Cons. | +Intg. | NP | +Infc | +SA | +C-E | +TA |
|---|---|---|---|---|---|---|---|---|---|
| A | .7515 | .7582 | .7795 | **.7796** | .7809 | .8023 | .8241 | .8654 | **.9055** |
| D | .3370 | .3556 | .3590 | **.3613** | .3979 | .4287 | .4399 | .4415 | **.4701** |
| L | .3266 | .3267 | .3310 | **.3657** | .3655 | .3920 | .4184 | .4713 | **.5144** |
| A/D | .3841 | .3890 | .4099 | **.4118** | .4586 | .4834 | .5083 | .5820 | **.6578** |
| D/A | .2497 | .2671 | .2562 | **.2689** | .3850 | .3921 | .4020 | .4222 | **.4355** |
| D/L | .2312 | .2433 | **.2497** | .2492 | .3334 | .3332 | .3316 | .3300 | **.3373** |
| L/D | .1923 | .2019 | **.2057** | .2056 | .2435 | .2668 | .2876 | .3311 | **.3932** |

boosts, it is essential for guiding the Transitive Annealer, informing about the potential oversizing or undersizing of communities.

**Effectiveness of Transitive Annealer**. The Transitive Annealer modules significantly enhance performance across various aspects, as shown in Table 4: **+Infc** excludes candidate cliques that violate the interface energy criteria, effectively pruning irrelevant cliques to boost efficiency and community detection quality. **+SA** utilizes a simulated annealing algorithm to merge neighboring candidates, refining the candidate set to achieve a more optimal community structure. **+C-E** shows performance gains by leveraging the full interconnectivity of cliques. **+TA** indicates that the initial candidates may not always be the best choices, as it dynamically identifies better community centers.

## 6.3 ADAPTABILITY EVALUATION

Table 5 presents the adaptability comparison of CLANN across different community sizes and numbers. The results demonstrate superior performance of CLANN across all datasets. As the community size increases, all models' performance tends to decrease. However, CLANN's decrease is much less pronounced than other models, suggesting better adaptability to complex structures. The results also apply to the other two datasets in Table 11 and 12. CLANN offers a more nuanced and accurate representation of community structures, which could better capture the intricacies of community formation and evolution by balancing stored energy with interface energy.

Table 5: Adaptability comparison on Amazon dataset with different community sizes and numbers.

|  | Metric | C-GAN | Bespoke | SEAL | CLARE | NP | CLANN |
|---|---|---|---|---|---|---|---|
| A-1k | F1 | .3446 | .9644 | .8331 | .9086 | .8339 | **.9905** |
|  | Jacc. | .2097 | .9463 | .7472 | .8483 | .7927 | **.9905** |
|  | ONMI | .0000 | .9411 | .7467 | .8676 | .7426 | **.9905** |
| A-2k | F1 | .4160 | .9163 | .7026 | .9140 | .7508 | **.9601** |
|  | Jacc. | .3103 | .8879 | .5915 | .8661 | .6803 | **.9452** |
|  | ONMI | .1947 | .8936 | .6099 | .8787 | .6569 | **.9490** |
| A-3k | F1 | .5003 | .8794 | .4414 | .8853 | .6783 | **.9452** |
|  | Jacc. | .4023 | .8328 | .3490 | .8213 | .5848 | **.9203** |
|  | ONMI | .3386 | .8463 | .3151 | .8281 | .5707 | **.9306** |
| A-4k | F1 | .5551 | .8199 | .3866 | .8583 | .6214 | **.9177** |
|  | Jacc. | .4491 | .7607 | .2976 | .7841 | .5129 | **.8764** |
|  | ONMI | .4150 | .7714 | .2706 | .7895 | .4354 | **.8957** |
| A-5k | F1 | .6602 | .7298 | .2381 | .6927 | .5223 | **.8084** |
|  | Jacc. | .5635 | .6575 | .1623 | .5897 | .4241 | **.7428** |
|  | ONMI | .5694 | .6432 | .0700 | .5725 | .3529 | **.7418** |

## 6.4 EFFICIENCY EVALUATION

As shown in Table 6, the total and average number of steps taken to anneal candidates underlines the model's efficiency again. The annealing process is a fine-tuning mechanism, making it more adaptable and versatile. The fact that CLANN can efficiently anneal all candidates, regardless of the quantity, indicates its superior handling of candidate communities compared to models like CLARE and SEAL. These models often limit the number of candidates, which might hinder their ability to detect all possible communities accurately.

Table 6: **NP**: NP runtime, **Clq**: clique core number, **T/S-(avg)**: total/average annealing runtime/step.

|  | NP(s) | # Clq | T(s) | T-avg(s) | S | S-avg |
|---|---|---|---|---|---|---|
| A | 35.2 | 4,995 | 112.1 | 0.022 | 11267 | 2.26 |
| D | 78.5 | 16,990 | 291.3 | 0.017 | 17984 | 1.06 |
| L | 147.9 | 30,000 | 945.5 | 0.032 | 37574 | 1.25 |
| A/D | 38.1 | 29,040 | 717.0 | 0.025 | 42732 | 1.47 |
| D/A | 75.9 | 5,775 | 124.2 | 0.022 | 9037 | 1.56 |
| D/L | 109.8 | 30,000 | 453.5 | 0.015 | 20329 | 0.68 |
| L/D | 212.3 | 30,000 | 839.2 | 0.028 | 42082 | 1.40 |

In addition, we compare the runtime with different community sizes. As shown in Fig. 4, CLANN achieves the best performance with much greater efficiency. This efficiency comes from three points: (1) Preliminary Core Filter prevents unnecessary clique searching; (2) Nucleus Proposer only checks possible cliques rather than all nodes' k-hop ego-networks; (3) Transitive Annealer refine candidate structure on clique-wise operation. Fig. 10 shows the runtime of each module.

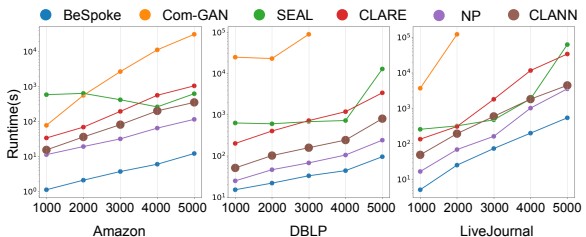

Figure 4: Runtime analysis. Nodes with bigger sizes stand for the best performance.

### 6.5 PARAMETER ANALYSIS

As illustrated in Fig. 5, we conduct a detailed loss weight analysis, where we first normalized the three loss items to the same magnitude. Then, for the target loss item, we varied its weight $\gamma^{\{E,C,I\}}$ across a range of values (0.01, 0.1, 1, 10, 100) to evaluate its influence on the final results. The analysis demonstrates that our model remains stable and robust across different loss weights.

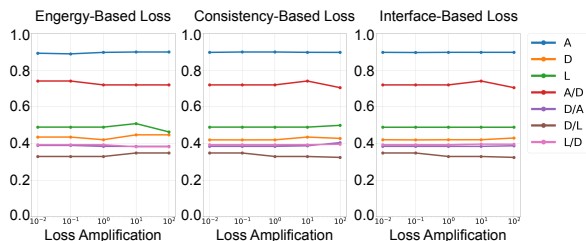

Figure 5: F1 scores of different loss weights.

### 6.6 CASE VISUALIZATION

We provide a detailed case analysis in Fig. 6. This analysis examines the annealing process across different community sizes. For small-sized communities, we may not need the annealing process, particularly when the community is a clique. In cases of larger community size, based on the clique-wise operation, CLANN can merge two cliques in a single step, thus converge efficiently. For large community, CLANN can also efficiently identify most of the core nodes, with errors typically occurring at the periphery. These peripheral errors usually involve nodes with either a single link to the main body or those that have multiple connections but do not truly belong to the core community.

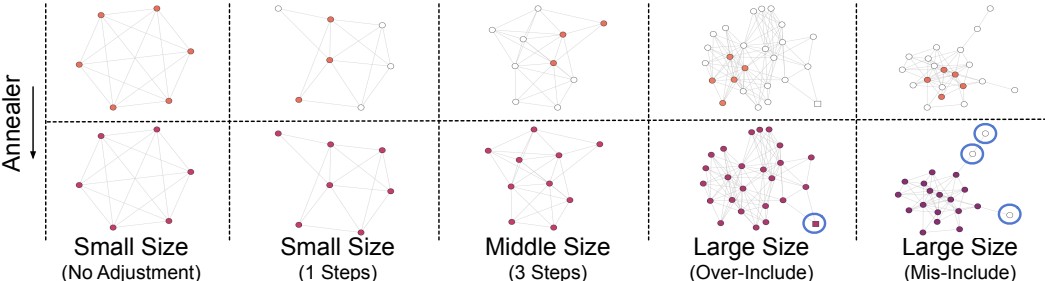

Figure 6: Case analysis on different community sizes. Our model takes only at most 3 steps to get the final communities for cases.

## 7 CONCLUSION AND FUTURE WORK

In this paper, we propose a novel community detection method CLique ANNealing (CLANN) inspired by the linking of the annealing process and community detection. Inconsistency in community cores and poor scalability are key challenges in semi-supervised community detection. To address these issues, CLANN introduces two main components: the Nucleus Proposer and the Transitive Annealer. The Nucleus Proposer enhances consistency between core candidates and actual community cores by incorporating crystallization kinetics into clique-based optimization. Meanwhile, the Transitive Annealer employs a learning-free growth process to boost scalability. Comprehensive evaluations highlight CLANN's superior performance, while also shedding light on the underlying mechanisms. Additionally, adaptability analysis underscores the model's applicability to real-world scenarios.

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

# A  CLIQUE-BASED MODULARITY AND LOSS IN COMMUNITY DETECTION

**Community Structure**: In all the datasets we used, the labeled communities inherently contain at least one clique. This aligns with the natural formation of communities, where tightly-knit groups of nodes (cliques) are common. Also, as shown in Fig. 7, all communities harbor at least one internal clique. Using these motifs (cliques) as starting points is thus more effective than traversing all nodes to locate suitable community centers Shen et al. (2009); Lu et al. (2010); Svendsen et al. (2015).

Besides, we design the homophily score in Fig. 8 to describe how central the nodes inside a community are. The score is calculated by using the number of neighbor nodes in the same community divided by the community size. We can see clique nodes have much higher homophily scores than those outside cliques.

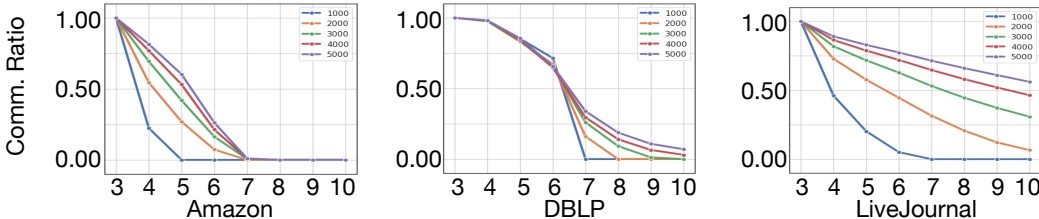

Figure 7: Accumulative proportion to community's max inner clique size.

**Modularity Consideration**: For the reason why cliques are used as the starting point, we can refer to modularity, which is calculated as the summation of the term $\left(a_{i,j} - \frac{k_i \cdot k_j}{2m}\right) \times \delta(i,j)$ over all pairs of nodes $i$ and $j$, where $a_{i,j}$ represents the actual connection between nodes $i$ and $j$, $k_i$ and $k_j$ are the degrees of nodes $i$ and $j$, and $m$ is the total number of edges in the graph. The $\delta(i,j)$ function is 1 if nodes $i$ and $j$ are in the same community, and 0 otherwise. In a clique, every pair of nodes is connected, making the term $\left(a_{i,j} - \frac{k_i \cdot k_j}{2m}\right)$ positive for all node pairs within the clique. This positive contribution is maximized when all nodes of the clique are treated as a single community, and breaking a clique into smaller communities reduces the overall modularity because it decreases the number of positive contributions in the summation.

**Energy-Based Loss**: To understand why the modularity of a clique with size $k$ is always greater than that of a sub-clique with size $k-1$, consider the modularity formula in detail. When you remove a node from a $k$-clique to form a $(k-1)$-clique, you also remove all the edges connected to that node. This reduces the number of positive terms in the modularity summation. Specifically, for each node pair $(i,j)$ within the original clique, the term $\left(a_{i,j} - \frac{k_i \cdot k_j}{2m}\right)$ contributes positively to the modularity when $i$ and $j$ are connected (which is always true in a clique). By removing a node from the community, you reduce the number of such positive contributions, thereby lowering the overall modularity. Therefore, the modularity of the original $k$-clique is always greater than that of any $(k-1)$-clique derived from it.

**Interface-Based Loss**: When expanding a community, the new additions must bring more benefit than the loss incurred. Referring to the term $\left(a_{i,j} - \frac{k_i \cdot k_j}{2m}\right)$, the first term $a_{i,j}$ is positive if there is an edge between nodes $i$ and $j$, but the second term $\frac{k_i \cdot k_j}{2m}$ is always negative because it subtracts from the overall modularity. This means that if the newly added node $j$ has strong connections to the existing community (resulting in more positive $a_{i,j}$ values), it can potentially overcome the negative contribution from the $\frac{k_i \cdot k_j}{2m}$ term, leading to an overall positive gain in modularity. Conversely, if most of the connections $a_{i,j}$ are zero, the gains from adding the node would not compensate for the loss, aligning the energy-based crystallization principle with modularity.

**Consistency-Based Loss**: Consistency is a natural property of well-structured communities. Better community structures tend to have tighter internal links, making it easier for a node's information to integrate into the community during GNN encoding. Particularly in cliques, all neighbors are one-hop neighbors of each other, ensuring strong consistency.

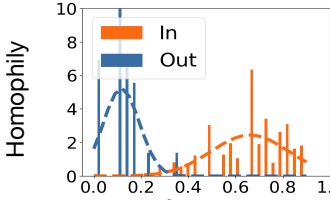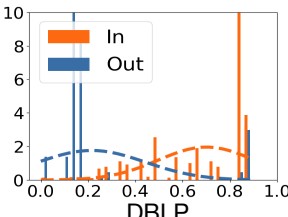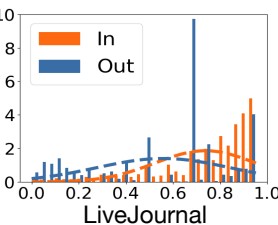

Figure 8: Homophily scores of community nodes inside (orange) and outside (blue) cliques.

**Integrity-Based Loss**: In practical scenarios, communities labeled by experts may not always align with the criterion of maximum modularity. Therefore, we introduce an additional loss to incorporate expert knowledge, recognizing that a community structure deemed appropriate by experts may not always correspond to the highest modularity score.

# B    EXPLANATION OF CRYSTALLIZATION KINETICS PRINCIPLES

If we compare a network to a crystal lattice structure, nodes signify atoms, and edges depict atomic bonds. Within an amorphous lattice structure, if we select and scrutinize a random area, we can find defects and deformations in it. This scenario mirrors constructing a subgraph from randomly chosen connected nodes, deemed a community. The so-called 'defects' or 'deformations' essentially represent mis-included or mis-excluded nodes within the community.

In the process of crystallization, annealing is a heat treatment that mitigates dislocations, repositions them into a configuration with lower energy, and promotes the formation of better grain boundaries. According to Rios et al. (2005), this process practically eradicates all dislocations prompted by deformation through the migration of grain boundaries. In the context of community detection, our proposed CLANN model emulates this annealing process. 'Defects' and 'deformations' within the network undergo 'annealing' to form well-structured communities by correctly including and excluding nodes. This process mirrors the organic adjustment of a crystal's structure, where misalignments are rectified, and a more coherent and unified formation emerges.

- **Stored Energy is Determined by the Grain Size and Defect Concentration**: In a crystalline structure, the material itself and those defects (such as dislocations, vacancies, and grain boundaries) store a significant portion of the energy. This energy is termed 'stored energy'. The size of the crystal grain and the concentration of these defects, therefore, largely influence the stored energy in the system. Also, the larger the grain, the higher the probability of containing defects, which leads to an increase in stored energy.

- **Crystallographic Directions Should be Consistent**: A well-crystallized crystal signifies that a material has a regular, repeating arrangement of atoms with minimal defects. As shown in Fig. 3, one characteristic of such a material is the consistency in crystallographic directions across grains, allowing for more uniform physical properties. On the other hand, if two crystal subgrains share the same crystallographic direction, they are very likely to form a more integrated grain. For instance, these consistent crystallographic directions can influence how the crystal behaves under stress or how it conducts heat or electricity.

- **Stored Energy and Boundary Interface Energy Barrier Decide the Crystal Growth**: Crystal growth depends on minimizing the material's total energy, which comprises the energy within the crystal itself and the energy at the interface or grain boundary. The driving force for the growth of a crystal subgrain is the reduction of energy density among the whole grain. If the energy barrier at the grain boundary is high, it might impede the grain's growth, even if there's a significant reduction in the energy within the crystal itself. Therefore, both the crystal's energy and the energy at the interface play a crucial role in determining whether the crystal will continue to grow.

- **Stored Energy and Defect Concentration Decide the Integrity**: For a selected area, if the area is full of well-crystallized crystals, we will get a larger average grain size and smaller defect concentration. Otherwise, there will be a smaller grain size and larger defect concentration, as there are more grain boundaries in this area. The comparison is shown in Fig. 3.

## C  COMPLEXITY ANALYSIS

Let $N$ represent the number of nodes and $M$ the specified number of cores. The entire model consists of three main components:

**Preliminary Core Filter**: This component identifies appropriate community center nodes using a simple Multi-Layer Perceptron (MLP) model. The training data is constructed based on betweenness centrality, but it is computed only on a few small sub-graphs, making the computation trivial and negligible compared to other parts of the model. Calculating the scores for all nodes to determine their suitability as community centers has a time complexity of $N$.

**Nucleus Proposer**: The primary time complexity of the Nucleus Proposer arises from clique preparation. Since the Nucleus Proposer only prepares cliques for specific nodes selected by the Core Filter, we thus only need to find the max cliques for these $M$ selected nodes. For each node, we need to find its neighbor nodes ($O(N)$) and check all edges among them ($O(N^2)$). The worst-case complexity is then $O(M \cdot (N + N^2))$, which simplifies to $O(N^2)$.

**Transitive Annealer**: As detailed in Algo. 2, each iteration of the Transitive Annealer involves 4 main steps (blue comments):

1. **Check Expandability**: Check the integrity score of the current state, thus $O(1)$.
2. **Collect Extendable Nodes**: Check all extendable neighboring nodes, in a worst-case $O(N)$.
3. **Check Nucleus Transition**: Check the transition condition, thus $O(1)$.
4. **Check Interface Requirement**: Similar to Step 2, in a worst-case $O(N)$.

If the maximum number of iterations is $C$, the total time complexity for Transitive Annealer is: $C(O(N) + O(N)) \rightarrow O(N)$. The total time complexity of CLANN is thus $O(N) + O(N) + O(N^2) \rightarrow O(N^2)$.

For runtime and convergence analysis, as illustrated in Fig. 4 and 10, CLANN's total runtime is better than most compared methods. The convergence behavior of the Transitive Annealer is shown in Table 6. In most cases, the Transitive Annealer converges within 3 steps. The case analysis in the appended PDF shows CLANN's behaviors under 5 settings with different community sizes, and CLANN will converge in at most 3 steps for all of them, which further justifies CLANN's scalability.

## D  PRELIMINARIES OF HYPERBOLIC GEOMETRY

Hyperbolic geometry encompasses several conformal models Cannon et al. (1997). Based on its widespread use in deep learning and computer vision, we operate on the Poincaré ball. The Poincaré ball is defined as $(\mathbb{D}_c^n, g^{\mathbb{D}_c^n})$, with manifold $\mathbb{D}_c^n = \{x \in \mathbb{R}^n : c\|x\| < 1\}$ and Riemannian metric:

$$g_x^{\mathbb{D}_c} = (\lambda_x^c)^2 g^E = \frac{2}{1 - c\|x\|^2} \mathbb{I}^n, \tag{14}$$

where $g^E = \mathbb{I}^n$ denotes the Euclidean metric tensor and $c$ is a hyperparameter governing the curvature and radius of the ball. Segmentation networks operate in Euclidean space and to be able to operate on the Poincaré ball, a mapping from the Euclidean tangent space to the hyperbolic space is required. The projection of a Euclidean vector $x$ onto the Poincaré ball is given by the exponential map with anchor $v$ and the Möbius addition $\oplus_c$:

$$\exp_v^c(x) = v \oplus_c (\tanh(\sqrt{c}\frac{\lambda_v^c\|x\|}{2})\frac{x}{\sqrt{c}\|x\|}),$$

$$v \oplus_c w = \frac{(1 + 2c\langle v, w\rangle + c\|w\|^2)v + (1 - c\|v\|^2)w}{1 + 2c\langle v, w\rangle + c^2\|v\|^2\|w\|^2}. \tag{15}$$

In practice, $v$ is commonly set to the origin, simplifying the exponential map to:

$$\exp_0^c(x) = \tanh(\sqrt{c}\|x\|)(x/(\sqrt{c}\|x\|)). \tag{16}$$

Besides, vector addition is not well-defined in the hyperbolic space (adding two points in the Poincaré ball might result in a point outside the ball). Instead, Möbius addition also provides an analog to

Table 7: Dataset statistics of community number $\#\hat{C}$, vertex number $\#V$, edge number $\#E$, max(average) community size $C_{M/A}$, the logarithm of edge number/vertex number $log(E/V)$, and coverage ratio $R_c$. $A$, $D$, and $L$ stand for Amazon, DBLP, and LiveJournal, respectively. Additionally, Est.$\alpha$ represents the estimated $\alpha$ value of degree power law fit.

|     | $\#\hat{C}$ | $\#V$ | $\#E$ | $C_{M/A}$ | $log(E/V)$ | $R_c$ | Est.$\alpha$ |
|-----|-------|-------|-------|-----------|-----------|-------|--------|
| A   | 1,000 | 6,926 | 17,893 | 30/9.4 | 1.37 | 0.812 | 12.91 |
| D   | 1,000 | 37,020 | 149,501 | 16/8.4 | 2.01 | 0.221 | 2.86 |
| L   | 1,000 | 69,860 | 911,179 | 30/13.0 | 3.71 | 0.169 | 3.21 |
| A+D | 2,000 | 43,946 | 172,394 | 30/8.9 | 1.97 | 0.128 | 2.95 |
| D+A | 2,000 | 43,946 | 172,394 | 30/8.9 | 1.97 | 0.186 | 2.95 |
| D+L | 2,000 | 106,880 | 1,070,680 | 30/10.7 | 3.32 | 0.077 | 3.30 |
| L+D | 2,000 | 106,880 | 1,070,680 | 30/10.7 | 3.32 | 0.111 | 3.30 |

Euclidean addition for hyperbolic space. Also, using hyperbolic embeddings, we should use the hyperbolic distance with the explicit formula:

$$d^c(x, y) = \frac{1}{\sqrt{|c|}}\text{acosh}\left(1 + \frac{2\|x - y\|^2}{(1 - \|x\|^2)(1 - \|y\|^2)}\right). \tag{17}$$

# E    DATASET STATISTICS

We utilize two dataset configurations. Initially, we adhere strictly to the data preparation and evaluation procedures of Wu et al. (2022); Zhang et al. (2020), involving three single datasets (Amazon(A), DBLP(D), and Livejournal(L)) and two hybrid datasets ("Amazon+DBLP"(A+D) and "DBLP+Livejournal"(D+L)). From a total of $5,000$ communities, communities exceeding the 90-th percentile size are excluded, and $1,000$ are randomly selected for experiments with 9%, 1%, and 90% designated as training, validation, and testing sets, respectively. For hybrid datasets, we introduce $5,000$ cross-network links between datasets like Amazon and DBLP, testing the model's ability to identify diverse community types. For instance, in the A/D setting, 90 Amazon communities are used for training, 10 for validation, and the rest for testing.

Additionally, to evaluate CLANN's adaptability to varying community sizes and numbers, each dataset is sorted by community size without excluding any community, forming 5 subsets such as "A-1k" for the smallest 1,000 communities to "A-5k" for all labeled communities. Split ratios are consistent with the first setting. Details of these arrangements are provided in Tab. 7 and Tab. 8.

# F    PRELIMINARY CORE FILTER

The Nucleus Proposer needs to prepare all nodes' cliques. However, for large graphs, finding all cliques is prohibitively time-consuming. To address this challenge, we develop a preliminary selection mechanism for large graphs before the Nucleus Proposer. Concretely, for each community, nodes with the highest betweenness centrality are labeled as community cores. The remaining nodes are considered peripheral nodes and labeled as $0$. To characterize the feature of node $v$, we concatenate the original representation $f_v^a$ and the average original features $\bar{f}_v^a$ of $v$'s neighbors. We train the preliminary classifier with communities from the training set:

$$y_v^c = \sigma(W^c[f_v^a || \bar{f}_v^a] + b^c), \tag{18}$$

where $y_v^c$ is the prediction for node $v$. $\sigma(\cdot)$, $W^c$, and $b^c$ are the activation function, weight parameters, and bias of the classifier. After feeding all nodes' features into the classifier, clique computations are exclusively performed for the top $M$ nodes and their neighbors in several hops. By employing this preliminary filter, we significantly reduce the time required for clique computation.

For betweenness calculation, while it is computationally expensive, we do not calculate it for all nodes. Instead, we calculate betweenness only for nodes within the training communities, which

Table 8: Dataset statistics of different community numbers (1k-5k). The meanings of each notation are identical to the previous table. Additionally, Est.$\alpha$ represents the estimated $\alpha$ value of degree power law fit.

|       | #V      | #E        | $C_{M/A}$     | log(E/V) | $R_c$ | Est.$\alpha$ |
|-------|---------|-----------|---------------|----------|-------|--------------|
| A-1k  | 1,032   | 1,156     | 4 / 3.3       | 0.11     | 0.767 | 26.41        |
| A-2k  | 2,819   | 4,596     | 7 / 4.3       | 0.49     | 0.776 | 17.50        |
| A-3k  | 5,401   | 10,878    | 10 / 5.6      | 0.70     | 0.790 | 19.98        |
| A-4k  | 9,478   | 22,522    | 18 / 7.5      | 0.87     | 0.800 | 8.83         |
| A-5k  | 19,905  | 54,618    | 328 / 13.5    | 1.01     | 0.840 | 4.48         |
| D-1k  | 26,027  | 88,945    | 6 / 6.0       | 1.23     | 0.224 | 3.32         |
| D-2k  | 47,416  | 181,396   | 7 / 6.3       | 1.34     | 0.254 | 3.04         |
| D-3k  | 70,529  | 280,695   | 9 / 6.8       | 1.38     | 0.270 | 3.24         |
| D-4k  | 97,435  | 397,563   | 12 / 7.6      | 1.41     | 0.288 | 3.30         |
| D-5k  | 216,556 | 829,388   | 7,556 / 22.4  | 1.34     | 0.431 | 6.65         |
| L-1k  | 10,252  | 25,724    | 6 / 4.2       | 0.92     | 0.356 | 5.59         |
| L-2k  | 37,967  | 274,146   | 13 / 6.7      | 1.98     | 0.293 | 2.44         |
| L-3k  | 100,435 | 1,097,204 | 21 / 10.0     | 2.39     | 0.234 | 2.56         |
| L-4k  | 234,820 | 3,401,944 | 35 / 14.3     | 2.67     | 0.177 | 4.03         |
| L-5k  | 439,450 | 7,431,647 | 1,441 / 27.8  | 2.83     | 0.192 | 2.69         |

comprise just 9% of the labeled communities. We extract each training community as a subgraph and only calculate betweenness within this subgraph. Subsequently, we use this neural network to identify community centers in the whole graph without needing to calculate betweenness for every node again, thus significantly reducing computational cost.

## G  EVALUATION METRICS & BASELINE

**Evaluation Metrics**. By convention, we select the bi-matching F1 and Jaccard scores Bakshi et al. (2018); Chakraborty et al. (2017); Jia et al. (2019); Zhang et al. (2020) as evaluation metrics. Given $N$ generated communities $\{\dot{C}^j\}$ and $M$ ground truth communities $\{\hat{C}^i\}$, scores are computed as:

$$\frac{1}{2}(\frac{1}{N}\sum_i \max_j \delta(\hat{C}^i, \dot{C}^j) + \frac{1}{M}\sum_j \max_i \delta(\dot{C}^j, \hat{C}^i)), \tag{19}$$

where $\delta(.,.)$ can be F1 or Jaccard function. Besides, we use the overlapping normalized mutual information (ONMI) McDaid et al. (2011) as a supplementary metric, the overlapping version of the NMI score. It is derived from the normalized mutual information, adjusted to ensure that it ranges between 0 and 1, where 0 indicates no correlation between the two community assignments, and 1 indicates a perfect match. For more information on ONMI, please refer to McDaid et al. (2011).

**Baselines**. We give details of our baseline methods of community detection:

- **BigClam** Yang & Leskovec (2013)[1] is designed for large-scale overlapping community detection. It recognizes densely connected overlaps between communities to enhance accuracy and scalability.

- **BigClam-A** Bakshi et al. (2018) stands for BigClam-Assisted, where BigClam is implemented on graphs modified by adding extra edges between nodes in the same community. These extra edges serve as additional constraints to the BigClam algorithm.

- **ComE** Cavallari et al. (2017)[2] leverages a synergistic loop between community and node embeddings to enhance graph visualization, community detection and node classification on multiple real-world datasets

---

[1]https://github.com/RobRomijnders/bigclam
[2]https://github.com/andompesta/ComE

- **CommunityGAN** Jia et al. (2019)[3] utilizes a Generative Adversarial Net (GAN) to generate the most likely motifs and optimize vertex embeddings, which indicate membership strength in communities.
- **vGraph** Sun et al. (2019)[4] is a probabilistic generative model that leverages a mixture model approach to represent nodes as combinations of communities.
- **Bespoke** Bakshi et al. (2018)[5] is a semi-supervised algorithm that leverages community membership information and node metadata to identify unique patterns in communities beyond traditional structures.
- **SEAL** Zhang et al. (2020)[6] uses a GAN to learn community detection heuristics from data, featuring a specialized GNN for generating communities and a seed selector for enhanced accuracy.
- **CLARE** Wu et al. (2022)[7] incorporates a Community Locator and Community Rewriter, utilizing deep reinforcement learning for community structure refinement.

## H    IMPLEMENTATION DETAILS

CLANN is implemented in Pytorch 2.1.0, PyG 2.4.0 with Python 3.9, and DeepSNAP 0.2.1. All experiments are conducted on AMD EPYC 7763 64-core Processor with 256GB of memory and a single NVIDIA RTX A5000 with 24GB of memory. In CLANN, the graph encoder is implemented by a 3-layer GCN with sum-pooling, and the hidden layers dimension $d$ is $64$ (identical to previous works), with a total of $104783$ (0.1M) parameters. The size of the model's checkpoint is 424 KB. Its weight parameters are optimized using Adam Kingma & Ba (2017) optimizer with 10 epochs and a learning rate of $1e^{-3}$ by default. $m$ in Tab. 1 is set to be 25, as we aim to maintain the structure with at least a smallest clique (k=3). For example, if we have a k=4 clique, by removing 25% of the nodes, we can still retain a k=3 clique. $\lambda_{clq}$ in Consistency-Based Loss is set to be 2. Loss coefficients $\gamma^{\{E,C,I\}}$ are set to keep each loss item in the same magnitude. The temperature probability function, $P_{\text{temp}}(|S_i^c|) = \Phi\left(\frac{|S_i^c| - \mu}{\sigma}\right)$, represents a normal distribution where $\mu$ is the mean and $\sigma$ is the standard deviation of the training community size.

Preliminary Core Filter is trained with training communities. For our largest dataset, lj-5k, it took 63.56 seconds for training and selection (betweenness is not calculated on the whole graph but on the community sub-graph, which usually only contains 30 nodes). All the competing methods are based on their publicly available official source code and are trained using the recommended optimization and hyperparameter settings in the original papers.

## I    SUPPLEMENTARY PARAMETERS ANALYSIS

**Candidate Size Rate**. As previously noted, the initial candidates generated by the Nucleus Proposer might not represent the optimal community centers. Therefore, we increased the candidate pool for the Transitive Annealer. Fig. 9 illustrates the comparison across various Candidate Size Rates. The model exhibits improved performance when annealing a broader set of candidates rather than solely relying on those provided by the Nucleus Proposer. In most instances, achieving superior community center detection necessitates annealing four times the number of candidates suggested by the Nucleus Proposer."

## J    MODULE RUNTIME ANALYSIS

We analyze the runtime performance of three different modules – Clique Prepare, Nucleus Proposer, and Transitive Annealer – across three separate datasets in Fig. 10. The datasets vary in size from $1,000$ to $5,000$ communities.

---

[3]https://github.com/SamJia/CommunityGAN
[4]https://github.com/sunfanyunn/vGraph
[5]https://github.com/yzhang1918/bespoke-sscd
[6]https://github.com/FDUDSDE/SEAL
[7]https://github.com/FDUDSDE/KDD2022CLARE

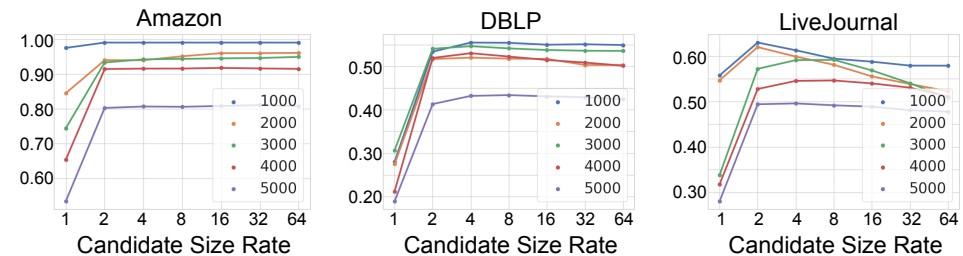

Figure 9: F1 performance of different candidate size rates and std rates.

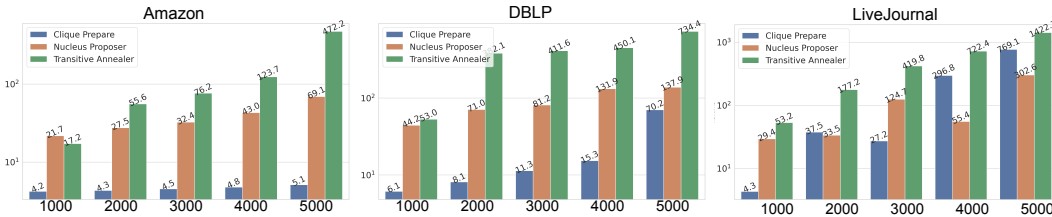

Figure 10: Runtime of different modules.

The runtime of the Clique Preparation module increases exponentially with the complexity of the graph. For instance, in the LiveJournal dataset, runtime rises from 1.57 seconds for 1,000 communities to 35,712.81 seconds for 5,000 communities, demonstrating substantial exponential growth as the community size and complexity increase. In contrast, the Nucleus Proposer module exhibits a more linear relationship with graph complexity. In the Amazon dataset, for example, runtime increases steadily from 7.09 seconds for 1,000 communities to 110.19 seconds for 5,000 communities, reflecting a consistent and predictable scaling with increasing community sizes.

The Transitive Annealer module's runtime initially increases exponentially with the complexity of the graph but tends to stabilize or converge at higher community numbers. For the LiveJournal dataset, runtime escalates from 32.28 seconds at 1,000 communities to 880.61 seconds at 5,000 communities, showing a tapering growth as it approaches larger datasets.

For smaller datasets, the major computational burden is attributed to the Transitive Annealer, where its runtime significantly surpasses that of other modules at initial community sizes. However, for larger real-world datasets, the Clique Preparation module becomes a significant bottleneck due to its exponential increase in runtime. In such scenarios, considering simpler motifs may be a promising choice to mitigate computational challenges and optimize performance.

## K  PERFORMANCE ON NON-CLIQUE, SPARSE, AND NOISY DATASETS

We evaluate CLANN's performance on non-clique structures, including bipartite graphs (3 collaboration and 3 co-purchase networks) and scatter-core networks as shown in Tables 16. To adapt CLANN for these graphs, cliques were replaced with scatter-based cores, resulting in CLANN(S). These experiments demonstrate that CLANN(S) maintains robust performance under non-clique and noisy conditions. The scatter-core approach effectively handles lower-density and overlapping community structures, addressing concerns regarding CLANN's reliance on cliques and extending its adaptability to diverse network types.

Additionally, as shown in Table 17, we conduct experiments on 15 datasets across three scenarios (Scatter-Core, Barabási-Albert, and NoisyDBLP datasets) to further evaluate CLANN(S)'s effectiveness in diverse settings. These scenarios simulate non-clique, scale-free, and noisy communities to assess the model's generalizability.

## K.1 ADAPTABILITY TO BIPARTITE NETWORKS

CLANN(S) shows significant improvement in bipartite networks by replacing clique-based cores with scatter cores. As shown in Table 16, CLANN(S) achieves the highest F1 scores across all datasets, including 0.5241 on CL1 and 0.5051 on CP2, outperforming CLARE and NP(S). This highlights CLANN(S)'s ability to identify community structures in networks lacking cliques, further validating its adaptability to non-clique environments.

## K.2 SCATTER-CORE NETWORKS (0 CLIQUES)

Scatter-core datasets were generated with 100,000 nodes distributed across varying community sizes (1K to 5K). For each community, nodes were connected as random tree structures to avoid clique, and inter-community edges were added while avoiding triangle formation. CLANN outperformed SOTA in all scatter-core datasets, achieving F1 scores up to 0.3011, highlighting its robustness in detecting coherent communities in sparse and minimally connected environments.

## K.3 BARABÁSI-ALBERT NETWORKS (<300 CLIQUES)

Barabási-Albert (BA) datasets, characterized by scale-free structures, were created with 100,000 nodes divided across varying community sizes (1K to 5K). High-degree nodes served as seeds, with connected nodes progressively added to maintain community growth. This process preserved the hierarchical and hub-dominated topology typical of BA networks. CLANN consistently outperformed SOTA, demonstrating its adaptability to scale-free structures.

## K.4 NOISYDBLP DATASETS

NoisyDBLP datasets were constructed by adding 10% noise (random edge additions and removals) to DBLP networks while preserving community connectivity. Across varying community sizes (1K to 5K), CLANN maintained stable performance, outperforming SOTA with F1 scores ranging from 0.4157 to 0.4742. This demonstrates the model's resilience to noise and overlapping structures.

The experimental results validate CLANN(S)'s generalizability and robustness across diverse network types, including scatter-core, bipartite, scale-free, and noisy graphs. By effectively adapting to different cores and maintaining strong performance in sparse, heterogeneous, and noisy environments, CLANN demonstrates its versatility as a robust community detection framework.

Table 9: Performance (Jaccard and ONMI) Comparisons with SOTA models.

| | Dataset | BigClam | BigClam-A | ComE | Com-GAN | vGraph | Bespoke | SEAL | CLARE | NP | CLANN |
|---|---|---|---|---|---|---|---|---|---|---|---|
| Jaccard | A | 0.5874 | 0.5623 | 0.5691 | 0.6045 | 0.5721 | 0.4415 | 0.6792 | 0.6827 | 0.7227 | **0.8600** |
| | D | 0.2186 | 0.2203 | N/A | 0.2830 | 0.0645 | 0.2593 | 0.2143 | 0.3132 | 0.3266 | **0.3703** |
| | L | 0.3102 | 0.3076 | N/A | 0.3183 | 0.0222 | 0.1324 | 0.3795 | 0.4027 | 0.3025 | **0.4382** |
| | A/D | 0.1102 | 0.1095 | N/A | 0.0109 | 0.0421 | 0.0488 | 0.2419 | 0.3241 | 0.4238 | **0.6247** |
| | D/A | 0.1485 | 0.1478 | N/A | 0.0610 | 0.0555 | 0.2135 | 0.0879 | 0.2166 | 0.3337 | **0.3601** |
| | D/L | 0.0523 | 0.0485 | N/A | 0.0120 | 0.0066 | 0.0756 | 0.1485 | 0.1893 | 0.2864 | **0.2893** |
| | L/D | 0.1505 | 0.1464 | N/A | 0.0097 | 0.0105 | 0.1907 | 0.1907 | 0.2308 | 0.1970 | **0.3356** |
| ONMI | A | 0.5865 | 0.5625 | 0.5570 | 0.6040 | 0.5532 | 0.4129 | 0.6862 | 0.7015 | 0.7404 | **0.8781** |
| | D | 0.1113 | 0.1110 | N/A | 0.2324 | 0.0020 | 0.2347 | 0.1603 | 0.2600 | 0.2799 | **0.3253** |
| | L | 0.2696 | 0.2641 | N/A | 0.3171 | <1e-4 | 0.1024 | 0.3695 | 0.3703 | 0.2768 | **0.4273** |
| | A/D | 0.0305 | 0.0334 | N/A | <1e-4 | <1e-4 | 0.0364 | 0.2475 | 0.3126 | 0.4261 | **0.6277** |
| | D/A | 0.0471 | 0.0477 | N/A | 0.0523 | <1e-4 | 0.1780 | 0.0380 | 0.1566 | 0.3108 | **0.3261** |
| | D/L | 0.0113 | 0.0065 | N/A | <1e-4 | <1e-4 | 0.0723 | 0.1155 | 0.1331 | 0.2648 | **0.2777** |
| | L/D | 0.0858 | 0.0795 | N/A | 0.0053 | <1e-4 | 0.1248 | 0.1906 | 0.2012 | 0.1808 | **0.3279** |

Table 10: Jaccard and ONMI Scores of different loss function and annealer schemes.

| | Dataset | $Engy.$ | $+Intf.$ | $+Cons.$ | $+Intg.$ | NP | +Infc | +SA | +C-E | +TA |
|---|---|---|---|---|---|---|---|---|---|---|
| Jaccard | A | 0.6925 | 0.7006 | 0.7228 | **0.7631** | 0.7227 | 0.7458 | 0.7696 | 0.8148 | **0.8600** |
| | D | 0.2749 | 0.2836 | 0.2932 | **0.2979** | 0.3266 | 0.3485 | 0.3452 | 0.3468 | **0.3703** |
| | L | 0.2590 | 0.2609 | 0.2660 | **0.3062** | 0.3025 | 0.3258 | 0.3491 | 0.3984 | **0.4382** |
| | A/D | 0.3520 | 0.3550 | 0.3839 | **0.3983** | 0.4238 | 0.4452 | 0.4666 | 0.5453 | **0.6247** |
| | D/A | 0.1955 | 0.2028 | 0.2107 | **0.2163** | 0.3337 | 0.3370 | 0.3391 | 0.3556 | **0.3601** |
| | D/L | 0.1762 | 0.1762 | **0.2008** | 0.1932 | 0.2864 | 0.2790 | 0.2817 | 0.2839 | **0.2893** |
| | L/D | 0.1495 | 0.1583 | **0.1607** | 0.1601 | 0.1970 | 0.2226 | 0.2508 | 0.2971 | **0.3356** |
| ONMI | A | 0.7075 | 0.7298 | 0.7411 | **0.7789** | 0.7404 | 0.7641 | 0.7887 | 0.8335 | **0.8781** |
| | D | 0.2454 | 0.2547 | 0.2610 | **0.2623** | 0.2799 | 0.3046 | 0.3110 | 0.3137 | **0.3253** |
| | L | 0.2355 | 0.2374 | 0.2401 | **0.2748** | 0.2768 | 0.3011 | 0.3254 | 0.3805 | **0.4273** |
| | A/D | 0.3517 | 0.3566 | 0.3801 | **0.4023** | 0.4261 | 0.4483 | 0.4705 | 0.5481 | **0.6277** |
| | D/A | 0.1597 | 0.1743 | 0.1717 | **0.1819** | 0.3108 | 0.3115 | 0.3115 | 0.3179 | **0.3261** |
| | D/L | 0.1476 | 0.1504 | 0.1596 | **0.1608** | 0.2684 | 0.2666 | 0.2677 | 0.2696 | **0.2777** |
| | L/D | 0.1319 | 0.1399 | 0.1446 | **0.1453** | 0.1808 | 0.2109 | 0.2787 | 0.3210 | **0.3279** |

Table 11: Adaptability comparison on DBLP dataset with different community sizes and numbers.

| Dataset | Metrics | Com-GAN | Bespoke | SEAL | CLARE | NP | CLANN |
|---|---|---|---|---|---|---|---|
| DBLP-1k | F1 | 0.1031 | 0.4259 | 0.0306 | 0.4755 | 0.2808 | **0.5547** |
| | Jaccard | 0.0822 | 0.3977 | 0.0181 | 0.3843 | 0.2061 | **0.5142** |
| | ONMI | 0.0665 | 0.3681 | 0.0024 | 0.3463 | 0.1335 | **0.4745** |
| DBLP-2k | F1 | 0.0823 | 0.4383 | 0.0877 | 0.4922 | 0.2749 | **0.5200** |
| | Jaccard | 0.0656 | 0.4096 | 0.0666 | 0.4011 | 0.2032 | **0.4701** |
| | ONMI | 0.0548 | 0.3794 | 0.0444 | 0.3609 | 0.1443 | **0.4326** |
| DBLP-3k | F1 | 0.0776 | 0.4380 | 0.1637 | 0.5105 | 0.3068 | **0.5464** |
| | Jaccard | 0.0606 | 0.4000 | 0.1048 | 0.4163 | 0.2337 | **0.4801** |
| | ONMI | 0.0494 | 0.3656 | 0.0256 | 0.3710 | 0.1924 | **0.4613** |
| DBLP-4k | F1 | N.A | 0.4187 | 0.4611 | 0.4993 | 0.2117 | **0.5343** |
| | Jaccard | N.A | 0.3736 | 0.3979 | 0.4008 | 0.1527 | **0.4608** |
| | ONMI | N.A | 0.3483 | 0.3934 | 0.3399 | 0.1173 | **0.4409** |
| DBLP-5k | F1 | N.A | 0.3193 | 0.2684 | 0.2893 | 0.1883 | **0.4688** |
| | Jaccard | N.A | 0.2744 | 0.2097 | 0.2246 | 0.1352 | **0.4064** |
| | ONMI | N.A | 0.2453 | 0.1948 | 0.1714 | 0.0981 | **0.3711** |

Table 12: Adaptability comparison on LiveJournal dataset with different community sizes and numbers.

| Dataset | Metrics | Com-GAN | Bespoke | SEAL | CLARE | NP | CLANN |
|---|---|---|---|---|---|---|---|
| LiveJournal-1k | F1 | 0.2922 | 0.4266 | 0.4193 | 0.5614 | 0.5563 | **0.6297** |
| | Jaccard | 0.2192 | 0.3532 | 0.3470 | 0.4561 | 0.4820 | **0.5512** |
| | ONMI | 0.1728 | 0.3051 | 0.3164 | 0.4426 | 0.4545 | **0.5457** |
| LiveJournal-2k | F1 | 0.1442 | 0.4312 | 0.3182 | 0.5547 | 0.5513 | **0.5916** |
| | Jaccard | 0.1175 | 0.3687 | 0.2391 | 0.4587 | 0.4752 | **0.5075** |
| | ONMI | 0.1076 | 0.3441 | 0.1654 | 0.4525 | 0.4675 | **0.4999** |
| LiveJournal-3k | F1 | N.A | 0.3903 | 0.2497 | 0.5480 | 0.3378 | **0.5717** |
| | Jaccard | N.A | 0.3331 | 0.1818 | 0.4557 | 0.2830 | **0.4867** |
| | ONMI | N.A | 0.3160 | 0.1225 | 0.4345 | 0.2672 | **0.4840** |
| LiveJournal-4k | F1 | N.A | 0.4099 | 0.1701 | 0.5224 | 0.3169 | **0.5462** |
| | Jaccard | N.A | 0.3497 | 0.1159 | 0.4275 | 0.2697 | **0.4580** |
| | ONMI | N.A | 0.3335 | 0.0548 | 0.3969 | 0.2606 | **0.4470** |
| LiveJournal-5k | F1 | N.A | 0.4298 | 0.1744 | 0.4350 | 0.2804 | **0.4758** |
| | Jaccard | N.A | 0.3634 | 0.1210 | 0.3460 | 0.2402 | **0.3904** |
| | ONMI | N.A | 0.3286 | 0.0644 | 0.3043 | 0.2398 | **0.3724** |

Table 13: F1 Score compared with unsupervised methods under Setting 1. For Spectral clustering methods, we present the best results among DBSCAN, HDBSCAN, and OPTICS. Additionally, Est. $\alpha$ represents the estimated $\alpha$ value of degree power law fit. N/A: not converge in 2 days.

| | Est.$\alpha$ | SBM | N-SBM | O-SBM | Louvain | Label Prop. | Spectral | CLANN |
|---|---|---|---|---|---|---|---|---|
| A | 12.91 | 0.3058 | 0.0371 | 0.0319 | 0.8226 | 0.7789 | 0.8226 | **0.9055** |
| D | 2.86 | 0.0924 | 0.0000 | 0.0068 | 0.1986 | 0.3777 | 0.3070 | **0.4701** |
| L | 3.21 | 0.1601 | 0.0000 | N/A | 0.4201 | 0.4801 | 0.4806 | **0.5144** |
| A/D | 2.95 | 0.0847 | 0.0371 | 0.0055 | 0.1206 | 0.4710 | 0.1575 | **0.6578** |
| D/A | 2.95 | 0.0876 | 0.0000 | 0.0057 | 0.1000 | 0.3606 | 0.1400 | **0.4355** |
| D/L | 3.30 | 0.0332 | 0.0000 | N/A | 0.0599 | 0.3346 | 0.0946 | **0.3373** |
| L/D | 3.30 | 0.1317 | 0.0000 | N/A | 0.2224 | 0.3849 | 0.3699 | **0.3932** |

Table 14: Runtime (in seconds) comparison with unsupervised methods under Setting 1.

| Dataset | SBM | N-SBM | Louvain | Label Prop. | DBSCAN | HDBSCAN | OPTICS | CLANN |
|---|---|---|---|---|---|---|---|---|
| A | 4.4 | 4.5 | 0.3 | 0.2 | 0.1 | 0.2 | 5.1 | 112.1 |
| D | 23.3 | 109.1 | 3.2 | 1.9 | 1.5 | 3.0 | 51.4 | 291.3 |
| L | 190.2 | 605.7 | 15.8 | 7.0 | 6.0 | 4.9 | 147.7 | 945.5 |
| A/D | 30.2 | 132.5 | 4.6 | 2.2 | 2.1 | 3.7 | 70.3 | 717.0 |
| D/A | 34.2 | 190.4 | 3.8 | 2.4 | 2.0 | 3.6 | 70.2 | 124.2 |
| D/L | 224.9 | 856.9 | 20.9 | 11.0 | 19.2 | 8.8 | 314.5 | 453.5 |
| L/D | 261.3 | 863.3 | 17.4 | 9.5 | 19.1 | 12.6 | 318.6 | 839.2 |

Table 15: F1 Score compared with top-3 unsupervised methods under Setting 2. Lv:Louvain, Lp: Label Propagation, Sp: Spectral, CL: CLANN.

| | Est.$\alpha$ | Lv | Lp | Sp | CL | | Est.$\alpha$ | Lv | Lp | Sp | CL | | Est.$\alpha$ | Lv | Lp | Sp | CL |
|---|---|---|---|---|---|---|---|---|---|---|---|---|---|---|---|---|---|
| A1 | 26.41 | .8546 | .8609 | .8546 | **.9905** | D1 | 3.32 | .2160 | .3976 | .3322 | **.5547** | L1 | 5.59 | .4334 | .5579 | .4713 | **.6297** |
| A2 | 17.50 | .8538 | .8612 | .8538 | **.9601** | D2 | 3.04 | .2185 | .4246 | .3521 | **.5200** | L2 | 2.44 | .4416 | .5661 | .5192 | **.5916** |
| A3 | 19.98 | .8391 | .8369 | .8391 | **.9452** | D3 | 3.24 | .1952 | .4333 | .3451 | **.5464** | L3 | 2.56 | .4384 | .5624 | .5232 | **.5717** |
| A4 | 8.83 | .8230 | .8271 | .8230 | **.9177** | D4 | 3.30 | .1811 | .4303 | .3631 | **.5343** | L4 | 4.03 | .4258 | .5249 | N/A | **.5462** |
| A5 | 4.48 | .7997 | .7237 | .7989 | **.8084** | D5 | 6.65 | .1839 | .3697 | N/A | **.4688** | L5 | 2.69 | .4438 | .4747 | N/A | **.4758** |

Table 16: F1 Score on general and bipartite graphs. Original CLANN can't be implemented on bipartite network, we build NP(S) and CLANN(S) by changing the core from clique to scatter (S). CL is collaboration network, where nodes are scientists and research papers. CP is co-purchase network, where modes are customers and products.

| | A (F1) | D (F1) | L (F1) | CL1 | CL2 | CL3 | CP1 | CP2 | CP3 |
|---|---|---|---|---|---|---|---|---|---|
| CLARE | 0.7730 | 0.3835 | 0.4950 | 0.4278 | 0.3612 | 0.4278 | 0.2158 | 0.2228 | 0.2017 |
| NP | 0.7809 | 0.3979 | 0.3655 | N/A | N/A | N/A | N/A | N/A | N/A |
| CLANN | **0.9055** | 0.4701 | 0.5144 | N/A | N/A | N/A | N/A | N/A | N/A |
| NP (S) | 0.6306 | 0.4615 | 0.4349 | 0.4190 | 0.3683 | 0.3500 | 0.3814 | 0.4312 | 0.3341 |
| CLANN (S) | 0.8933 | **0.4739** | **0.5154** | **0.5241** | **0.4477** | **0.5494** | **0.4312** | **0.5051** | **0.4923** |

Table 17: Robustness of CLANN in Diverse Network Environments: F1 Performance on Scatter, Barabási-Albert, and NoisyDBLP Datasets with Scaling Community Structures (1K-5K Communities)

| Scatter Datasets | | | | | |
|---|---|---|---|---|---|
| # Community | 1K | 2K | 3K | 4K | 5K |
| # Node | 64,020 | 100,000 | 100,000 | 100,000 | 100,000 |
| # Edge | 66,665 | 200,000 | 200,000 | 200,000 | 200,000 |
| CLARE | 0.1929 | 0.1522 | 0.1904 | 0.2153 | 0.2394 |
| CLANN | **0.3100** | **0.1567** | **0.2824** | **0.2980** | **0.3011** |
| Barabási-Albert Datasets | | | | | |
| # Community | 1K | 2K | 3K | 4K | 5K |
| # Node | 100,000 | 100,000 | 100,000 | 100,000 | 100,000 |
| # Edge | 200,000 | 200,000 | 200,000 | 200,000 | 200,000 |
| CLARE | 0.1291 | 0.1871 | 0.2160 | 0.2458 | 0.2589 |
| CLANN | **0.1922** | **0.2513** | **0.2992** | **0.3300** | **0.3307** |
| NoisyDBLP Datasets | | | | | |
| # Community | 1K | 2K | 3K | 4K | 5K |
| # Node | 25,968 | 47,318 | 70,425 | 97,270 | 215,912 |
| # Edge | 88,949 | 181,547 | 280,586 | 397,201 | 829,877 |
| CLARE | 0.3308 | 0.3608 | 0.3660 | 0.3753 | 0.2837 |
| CLANN | **0.4157** | **0.4475** | **0.4742** | **0.4398** | **0.3539** |

---

**Algorithm 1:** Nucleus Proposer

---

**Input** : Graph $G$, Training communities $C$, Recorded Clique Set $Q$, Max Epoch $E_M$, Loss Contribution Coefficients $\gamma^{\{E,C,I\}}$, Learning Rate $\alpha$, Initial State Number $M$.

**Output :** Initial State $S_{init}$, Graph Encoder $H$, Status Classifier Parameters $W_p^k$, $b_p^k$ (in Eq. 6)

1 Epoch: $e \leftarrow 1$, $S_{init} \leftarrow \{\}$;

2 **while** $e < E_M$ **do**

3    // Prepare positive and negative batches

4    Sample $a_1, a_2, a_3$ from $C$;

5    Sample $b, c$ from $a_1$;

6    $S \leftarrow \{a_1, a_2, a_3, b, c\}$;

7    // Calculate Energy, Consistency, and Interface losses

8    $loss^E \leftarrow loss_{Size}^E(S) + loss_{Defc}^E(S)$;

9    $loss^C \leftarrow loss^C(S)$;

10   $loss^I \leftarrow loss^I(S)$;

11   // Update encoder

12   $H := H - \alpha \bigtriangledown (\gamma^E loss^E + \gamma^C loss^C + \gamma^I loss^I)$;

13 **while** $e < E_M$ **do**

14   // Prepare positive and negative batches

15   Sample $a_1, a_2, a_3$ from $C$;

16   Sample $b, c$ from $a_1$;

17   $S \leftarrow \{a_1, a_2, a_3, b, c\}$;

18   // Calculate Integrity loss

19   $loss^G \leftarrow loss^G(S)$;

20   // Update Status Classifier

21   $W_p^k, b_p^k := W_p^k, b_p^k - \alpha \bigtriangledown loss^G$;

22 // Select initial states

23 Clique Embedding Set $h_Q \leftarrow H(Q)$;

24 **for** $c \in C$ **do**

25   Community Embedding $h_c \leftarrow H(c)$;

26   Embedding Distance $d_{c,Q} \leftarrow ||h_c - h_Q||$;

27   Sort $d_{c,Q}$ with ascending order;

28   Append first $\lfloor M/|C| \rfloor$ Cliques into $S_{init}$;

29 **return** $S_{init}$, $H$

---

---

**Algorithm 2:** Transitive Annealer

---

**Input** : Initial state $S_{init}$, Max Step $M$.

**Output :** Annealed community $\hat{S}$.

1 Ending Flag: $F_{end} \leftarrow False$;

2 **while** $F_{end} \neq True$ **do**

3     // Check Expandability

4     $\hat{S} \leftarrow S_{init}$;

5     $\hat{y}_{S_{init}} \leftarrow$ Integrity score of $S_{init}$;

6     **if** $\hat{y}^1_{S_{init}} < max(\hat{y}_{S_{init}})$ **then**

7         Break ;

8     // Collect Extendable Nodes and Calculate Properties

9     Extendable nodes set $V^e = \{v^e_1, \ldots, v^e_{|V^e|}\}$;

10     Extendable Node Properties $E_{ex} \leftarrow \{\}$;

11     **for** $v^e_i \in V^e$ **do**

12         $C^e_i \leftarrow$ merge all cliques of $v^e_i$ and exclude nodes in $S_{init}$;

13         Calculate properties and append to $E_{ex}$;

14     // Check Nucleus Transition

15     $k \leftarrow$ max(Integrity Scores);

16     **if** $\hat{y}^2_{S_{init}} < \hat{y}^2_{C^e_i}$ **then**

17         $S_{init} \leftarrow C^e_k$;

18         Continue ;

19     // Check Energy Score and Interface Requirement

20     Node score $P_{ex} \leftarrow$ Softmax(Norm Difference List);

21     **for** $i \leftarrow 1$ *to* $|V^e|$ **do**

22         Energy Flag $F_E \leftarrow P_{ex}[i] \geq P_{temp}(|\hat{S} \cup C^e_i|)$);

23         Interface Flag $F_I \leftarrow$ Interface Energy Check;

24         **if** $F_E$ *and* $F_I$ **then**

25             // Update State

26             $\hat{S} \leftarrow \hat{S} \cup C^e_i$;

27     $M \leftarrow M - 1$;

28     **if** $\hat{S} = S_{init}$ *or* $M \leq 0$ **then**

29         $F_{end} \leftarrow True$;

30 **return** $\hat{S}$

---

