# OpenReview forum: "New Recipe for Semi-supervised Community Detection: Clique Annealing under Crystallization Kinetics"
_ICLR.cc/2025/Conference — Submitted to ICLR 2025_

### Official Review · Reviewer_oMJD · 2024-10-30

**Soundness:** 3
**Presentation:** 2
**Contribution:** 3
**Rating:** 6
**Confidence:** 4

**Summary:**

This manuscript introduces CLANN, a new community detection method motivated by the kinetics crystallization process. It integrates the Nucleus Proposer to learn to detect community cores and Transitive Annealer for spontaneous growth. Extensive experiments with various settings highlight the excellence of CLANN.

**Strengths:**

S1: The design of the method is inspired by the kinetics of the crystallization process, which provides strong motivation.

S2: Detailed formulas are used to express the approach more rigorously and scientifically.

S3: Extensive experiments are conducted to validate the method.

**Weaknesses:**

W1: I have two key concerns about the design of the loss functions

W1.1: In equation 7, the formulation of Integrity-Based Loss is used to guide the prediction of the state of undergrown, equilibrium, and overgrown states. It is modeled as a classification problem. However, the loss function used seems not to align with the Binary Cross Entropy.

The binary cross-entropy loss is:

$
\text{BCE}(y, \hat{y}) = - \frac{1}{N} \sum_{i=1}^N \left( y_i \log(\hat{y}_i) + (1 - y_i) \log(1 - \hat{y}_i) \right)
$

where:
- \( y_i \) is the true label (0 or 1),
- \( \hat{y}_i \) is the predicted probability of the positive class,
- \( N \) is the total number of samples.

while the authors seem to mix up the order in the bracket:

$
\sum_{g=1}^3 \left( (y_g^k - 1) \log(\hat{y}_g^k) - y_g^k \log(1 - \hat{y}_g^k) \right)
$

W1.2: In the design of the integrity-based method, the overflow states are predefined, which limits the inclusion of many promising patterns, such as:
$(a_2 + \check{a}_2)$

===============

W2: I have two key concerns about the training process.

W2.1: A two-stage training method is used here: first, the Energy-Based Loss, Consistency-Based Loss, and Interface-Based Loss are trained together, followed by training the Integrity-Based Loss separately. This approach to multi-loss training is relatively unusual and lacks clear motivation. Won't the second loss disrupt the model trained in the first stage? How are these losses aligned?

W2.2: As seen in Figure 5, when Loss Amplification approaches 0, performance on many datasets actually improves, indicating that this loss does not enhance training and may even reduce performance (e.g., Consistency-Based Loss and Interface-Based Loss). Strangely, in the ablation study in Table 4, these losses appear to be beneficial.

===============

W3: Some claims and typos can be improved.

W3.1: The authors argue that previous reinforcement-learning-based methods face scalability issues (Line 017), and that the proposed method has better scalability compared to these approaches. CLARE itself is also a reinforcement learning-based method (Line 984). As shown in Figure 4, NP and CLANN demonstrate poorer scalability compared to CLARE. In the DBLP and LiveJournal datasets, NP and CLANN even exhibit faster growth in runtime.

W3.2: Line 157, "Propose" => "Proposer"

**Questions:**

Given that some losses do not statistically significantly improve performance (e.g., Integrity-Based Loss), would it be possible to reduce certain losses to make the model simpler and emphasize the more effective components?

---

> ### Author Response · Authors · 2024-11-23
>
> > ## W1.1+Q2. Loss function order mix-up and typo.
>
> We appreciate your observation regarding the order mix-up error. We have rectified the error and the typo in the revised manuscript with blue color.
>
> > ## W1.2. Concern about missing promising patterns.
>
> Thanks for your thoughtful feedback on the predefined overflow states and their potential limitations in accommodating promising patterns. Below, we provide clarification to address this concern.
>
> 1. Definition of Overflow States:
> The **undergrown**, **equilibrium**, and **overgrown** states are not arbitrarily predefined by our method. Instead, they are derived directly from the **labeled communities**, which represent expert-labeled classifications of community growth dynamics and are therefore grounded in the structure and semantics of the labeled dataset. By using these states as references, the model aligns well with real-world patterns observed in the training data.
>
> 2. Role of the Integrity-Based Loss:
> The **Integrity-Based Loss** leverages these labeled states to train a classifier that predicts the growth status of subgraphs. This approach ensures that the model's predictions are consistent with the expert-labeled community structures and allows for robust guidance during the **Transitive Annealer** process.
>
> 3. Handling Patterns like $a + \breve{a}$:
> Suppose **node-1** is the community core of $a$, while **node-2** is the core of a community that covers $a + \breve{a}$. For the growth process starting from **node-1**, $a + \breve{a}$ can be classified as an overgrown state. However, for growth starting from **node-2**, it is considered as an equilibrium state, our model can thus include these promising candidates by considering different start points during the annealing process.
>
> > ## W2.1. Two-stage training scheme.
>
> After training the **Energy-Based**, **Consistency-Based**, and **Interface-Based** loss functions, we fix the graph encoder parameters. The **Integrity-Based Loss** training does not disrupt the learned representations. The **Integrity-Based Loss** is specifically designed to optimize a separate classifier that takes subgraph embeddings as input and outputs their growth status. This loss is not directly connected to the optimization of the graph encoder but instead focuses on facilitating decision-making during the **Transitive Annealer** stage. Training the **Integrity-Based Loss** separately can avoid additional constraints or conflicting objectives on the graph encoder during the first stage.
>
> > ## W2.2+Q1. Loss Contribution.
>
> Thank you for highlighting this point. The interaction of the three losses during joint training contributes to the observed result. These losses interact dynamically, and their combined contribution to the final performance is not a linear sum of individual ones. When the three losses are trained together, adjustments to one loss's weight can lead to performance fluctuations due to these interactions, we thus can hardly conclude that one loss function is entirely invalid or dominant.
>
> As mentioned before, the **Integrity-Based Loss**, while not directly optimizing representation quality, serves a specialized role in guiding the stopping criteria during the growth process of the **Transitive Annealer**. This loss ensures robust and consistent growth of community structures by providing a clear indicator for terminating growth states. Its contribution lies in improving interpretability and the overall scalability of the annealing process, rather than directly enhancing network representation. Removing or simplifying these losses may inadvertently compromise this robustness, particularly in datasets with diverse community characteristics.
>
>
> > ## W3.1. Statement about superior scalability.
>
> Thank you for highlighting this issue. Since the **Clique Preparation** phase is the most time-intensive part of CLANN, we focused on optimizing its efficiency. After thoroughly reviewing the code, we introduced a preprocessing step to eliminate unnecessary operations, significantly enhancing the efficiency of the clique preparation module. We also replaced the result in the revised version. The new source code can be found in an anonymous GitHub repository at https://anonymous.4open.science/r/CLANN_ICLR2025-51D0/README.md.
>
> Here we provide the runtime improvement based on our new code:
>
> ```
> Table 1. Runtime Improvement
> ```
>
> | Dataset | Size | Old Runtime | New Runtime | Change |
> |-|-|-|-|-|
> | **Amazon** | 1K | 0.54 | 4.21 | +3.67 |
> |  | 2K | 0.84 | 4.33 | +3.49 |
> |  | 3K | 1.15 | 4.54 | +3.39 |
> |  | 4K | 1.24 | 4.77 | +3.53 |
> |  | 5K | 2.96 | 5.12 | +2.16 |
> | **DBLP** | 1K | 4.47 | 6.12 | +1.65 |
> |  | 2K | 9.15 | 8.12 | -1.03 |
> |  | 3K | 17.58 | 11.35 | -6.23 |
> |  | 4K | 29.76 | 15.32 | -14.44 |
> |  | 5K | 272.92 | 70.21 | -202.71 |
> | **LJ** | 1K | 1.57 | 4.32 | +2.75 |
> |  | 2K | 15.73 | 37.54 | +21.81 |
> |  | 3K | 290.38 | 27.21 | -263.17 |
> |  | 4K | 3996.50 | 296.78 | -3699.72 |
> |  | 5K | 35712.81 | 769.10 | -34943.71 |

---

> > ### Comment · Reviewer_oMJD · 2024-11-26
> >
> > Thank you very much for your rebuttal and the improvements addressing the issues in W1, W2.1, and W3. These revisions have significantly enhanced the quality of the paper.
> >
> > Regarding W2.2: As shown in Figure 5, when Loss Amplification approaches 0, performance on many datasets actually improves. This suggests that the loss does not effectively enhance training and may even degrade performance (e.g., Consistency-Based Loss and Interface-Based Loss). However, in the ablation study presented in Table 4, these losses appear to contribute positively, which creates a discrepancy.
> >
> >
> > The authors have provided only a high-level explanation of the role of these loss functions, without offering theoretical or experimental evidence to support their claims (e.g., "adjustments to one loss's weight can lead to performance fluctuations due to these interactions", The author did not elaborate on the underlying relationships but merely provided a superficial explanation, which might be insufficient for an ICLR research paper.). This lack of substantiation reduces the persuasiveness of the argument and raises concerns about whether these loss functions are effectively contributing to the model’s performance.
> >
> >
> > Overall, this is a nice and well-motivated piece of work. Overall, considering the above points, I will maintain a rating of 6.

---

> > > ### Author Response · Authors · 2024-11-26
> > >
> > > Thank you for your constructive comments and for acknowledging the improvements in the revised version of our manuscript. We appreciate your detailed feedback, especially on the interactions among the loss functions and their contributions to the model.
> > >
> > > Regarding the interaction between the loss functions, we acknowledge the limitations in providing theoretical or experimental evidence to fully substantiate our explanation. As you noted, there might indeed be overlaps or redundancies among the designed loss functions. Our intention behind incorporating these multiple losses was to emulate the multi-faceted nature of the physical crystallization process and to map these principles metaphorically to community detection.
> > >
> > > Moreover, as our task involves semi-supervised community detection, we recognized that there are inherent discrepancies between the human-labeled communities and those derived purely from natural energy-based formations. To bridge this gap, we included these additional loss components to align the generated communities more closely with expert-labeled structures.
> > >
> > > However, we agree that a rigorous theoretical analysis and more detailed experimental validations are essential to fully understand the interactions and individual contributions of these loss components. In future work, we plan to start from unsupervised community detection tasks, where we will conduct a comprehensive theoretical analysis and experimental study to isolate and evaluate the impact of these principles. Additionally, we will explore how expert-provided labels influence the interaction between these loss functions to replicate these principles effectively.
> > >
> > > Once again, we are deeply grateful for your time, your recognition of the strengths of our work, and your invaluable suggestions. Your feedback has significantly contributed to improving the quality and clarity of our manuscript!

---

> ### Comment · Area_Chair_fmVn · 2024-11-25
>
> Could please acknowledge and respond to the rebuttal.

---

### Official Review · Reviewer_pfE6 · 2024-11-03

**Soundness:** 2
**Presentation:** 2
**Contribution:** 2
**Rating:** 5
**Confidence:** 3

**Summary:**

The paper proposes a novel approach for semi-supervised community detection, named CLique ANNealing (CLANN), which integrates principles from crystallization kinetics with community detection. To address the issues with scalability and core selection inconsistency, CLANN introduces two main components Nucleus Proposer and Transitive Annealer, draws an analogy between community detection and crystallization, treating communities as grains growing from smaller, well-defined "clique" cores. Experiments conducted on datasets show that CLANN outperforms state-of-the-art methods in both efficacy and efficiency.

**Strengths:**

S1: The approach of using crystallization principles, especially the spontaneous annealing process, provides a compelling and physics-informed basis for community formation.
S2: The learning-free Transitive Annealer circumvents computational issues tied to reinforcement learning, showing superior scalability over existing methods.
S3: Empirical results indicate that CLANN consistently achieves higher performance across diverse datasets, particularly excelling in detecting distinct community structures within hybrid graphs.

**Weaknesses:**

W1: The CLANN model assumes high network density and consistency by analogizing community detection to crystallization growth. In practice, however, many graph networks (e.g., social networks, literature citation networks, etc.) may contain highly heterogeneous connectivity structures and low degree of node density, which could lead to difficulties for the annealing process to efficiently identify core or stable structures. In sparse or non-uniform networks, the use of an annealing process may lead to difficulties in forming coherent communities in the model, affecting the detection accuracy.
W2: CLANN pre-screens the candidate set by a filtering mechanism when initializing candidate community cores to improve efficiency. However, on large-scale datasets, this mechanism still requires a lot of computational resources to traverse and filter the candidate cores. Especially in real-time application scenarios, the initialization process may be too slow, affecting the application efficiency of the algorithm.
W3: For hyperparameters such as Transitive Threshold $\theta$, Temperature Prob. $P_{temp}$, further analysis is required.

**Questions:**

Q1: Eq. 9 is not the same as the loss in Algo. 1? I think this needs more explanation. Also, at the same time the $ \gamma _G$ should be performing a parameter sensitivity analysis.

---

> ### Author Response · Authors · 2024-11-23
>
> > ## W1. The difficulties in dealing with heterogeneous connectivity structures and low degree of node density.
>
> Thanks for your constructive comments. To demonstrate the effectiveness of CLANN in handling heterogeneous structures and low node density, we first conducted experiments on non-clique structures, focusing specifically on bipartite graphs, as shown in **Appendix Section K**.
> By replacing cliques with scatter-based cores, we tested the model on **6** datasets (**3** collaboration and **3** co-purchase networks). These experiments further indicate that CLANN retains robust performance under non-clique and noisy network scenarios.
>
> Moreover, we conducted experiments on **15** datasets (**3** scenarios with **5** datasets in each) to further justify CLANN's (with scatter cores) effectiveness in diverse situations (**Non-Clique**, **Sparse**, and **Noisy**). Here we also provide the results of CLANN's performance on newly introduced **Non-Clique, Sparse, and Noisy** networks as shown below. The detailedresults and analysis can be found in the common response **Global Response 2**.
>
> ```
> Table 1. CLANN's F1 Performance Across 3 New Scenarios
> ```
>
> |Dataset|S 1K|S 2K|S 3K|S 4K|S 5K|B 1K|B 2K|B 3K|B 4K|B 5K|N 1K|N 2K|N 3K|N 4K|N 5K|
> |-|-|-|-|-|-|-|-|-|-|-|-|-|-|-|-|
> |#Node|64,020|100,000|100,000|100,000|100,000|100,000|100,000|100,000|100,000|100,000|25,968|47,318|70,425|97,270|215,912|
> |#Edge|66,665|200,000|200,000|200,000|200,000|200,000|200,000|200,000|200,000|200,000|88,949|181,547|280,586|397,201|829,877|
> |CLARE|0.1929|0.1522|0.1904|0.2153|0.2394|0.1291|0.1871|0.216|0.2458|0.2589|0.3308|0.3608|0.366|0.3753|0.2837|
> |CLANN|0.31|0.1567|0.2824|0.298|0.3011|0.1922|0.2513|0.2992|0.33|0.3307|0.4157|0.4475|0.4742|0.4398|0.3539|
>
>
> > ## W2. Traverse and filter the candidate cores may require large computational resources on large datasets.
>
> We appreciate the reviewer’s valuable feedback on the computational overhead of the candidate filtering mechanism, especially in large-scale datasets and real-time application scenarios. We would like to clarify the following points to address this concern:
>
> - **Efficiency of the Preliminary Core Filter**: The Preliminary Core Filter pre-selects candidate community cores by only leveraging a **lightweight** classifier (input vector dimension is only **5**). The standard augment feature for this filter only involves simple operations (**degree calculation**), which are computationally lightweight and can be performed in near real-time. This approach ensures the filtering mechanism remains efficient and scalable, even in real-time or streaming scenarios.
>
> - **Real-Time Adaptation for New Nodes**: For dynamic graphs where new nodes are added, the computational requirement is further minimized. When a new node comes in, instead of recomputing all existing nodes, we only update very basic standard augment feature for the related neighbor nodes (degree-based features, Page 6 Line 272-273), because the new node only connects to a few existing nodes, which will not change the standard augment feature of most unconnected nodes.
>
> > ## W3. More analysis on the hyperparameters
>
> Thanks for the constructive comments. In the real deployment, to improve model's generalizability, we do not need to set the annealing temperature threshold and transitive threshold manually. Please refer to our reply in **Global Response 3** in the Global Response section above for this concern.
>
> > ## Q1. More explanation on Eq.9 and sensitivity analysis on $\gamma_{G}$.
>
> **Reply**: We appreciate the reviewer’s observation regarding the inconsistency between **Eq. 9** and the loss function in **Algorithm 1**. Below, we clarify this issue and explain the reasoning behind our design choices.
>
> 1. Rectification of Algorithm 1:
> We acknowledge the inconsistency and have rectified **Algorithm 1** to align it with the explanation in the main text. Specifically, **$\gamma_{G}$**, which corresponds to the **Integrity-Based Loss**, has been removed from **Algorithm 1**. This adjustment reflects the fact that the **Integrity-Based Loss** is optimized independently and not during the main training process for the graph encoder.
>
> 2. Separate Optimization of the Integrity-Based Loss:
> As noted in our response to Reviewer **oMJD W2.1**, the **Integrity-Based Loss** is specifically designed to optimize a separate classifier. This classifier takes subgraph embeddings as input and outputs the growth status (undergrown, equilibrium, or overgrown). By optimizing this loss separately:
>   - We avoid introducing additional hyperparameters ($\gamma_{G}$) that would complicate the optimization of the graph encoder.
>   - The graph encoder focuses solely on optimizing the **Energy-Based Loss**, **Consistency-Based Loss**, and **Interface-Based Loss**, which are critical for learning robust subgraph embeddings.

---

> ### Comment · Area_Chair_fmVn · 2024-11-25
>
> Could please acknowledge and respond to the rebuttal.

---

> ### Author Response · Authors · 2024-11-27
>
> Dear Reviewer pfE6:
>
> We would like to express our sincere gratitude for your time in reviewing our paper and your valuable comments.
>
> Since the revision submission period is nearing its end, we are wondering whether our response has sufficiently addressed your questions. If so, we would greatly appreciate it if you could consider updating the score to reflect this. If you have any additional suggestions, we are more than willing to engage in further discussions and make necessary improvements to the paper.
>
> Thank you once again for dedicating your time to enhancing our paper!
>
> Best regards,
>
> Authors

---

> ### Author Response · Authors · 2024-12-01
>
> Dear Reviewer pfE6,
>
> Since the End of author/reviewer discussions is coming in ONE day, may we know if our response addresses your main concerns? If so, we kindly ask for your reconsideration of the score. Should you have any further advice on the paper and/or our rebuttal, please let us know and we will be more than happy to engage in more discussion and paper improvements.
>
> Thank you so much for devoting time to improving our paper!

---

### Official Review · Reviewer_hnzb · 2024-11-03

**Soundness:** 3
**Presentation:** 2
**Contribution:** 3
**Rating:** 6
**Confidence:** 2

**Summary:**

In community detection, semi-supervised methods are widely used to identify specific communities; however, existing methods often rely on high-cost models like Generative Adversarial Networks (GAN) or reinforcement learning, resulting in suboptimal initial community cores and limited scalability. There are two main issues with these methods: first, community core inconsistency, as growth-based methods often start from random nodes or k-hop ego networks, which do not accurately represent the structural characteristics of the community; and second, poor growth scalability, as using GAN or reinforcement learning to increase community core candidates is computationally intensive and fails to efficiently leverage existing information, limiting the number of core candidates. To address these issues, the authors propose a community detection model, CLANN, based on annealing and crystallization kinetics. This model comprises two key components: the Nucleus Proposer, which selects completely connected subgraphs (i.e., cliques) as community cores and integrates four crystallization principles to enhance core consistency; and the Transitive Annealer, which uses a learning-free annealing mechanism to merge adjacent cliques into final communities, thereby reducing computational cost and improving scalability. Experimental results show that CLANN outperforms existing community detection methods in terms of accuracy and efficiency across multiple real-world datasets, demonstrating its effectiveness and adaptability in diverse network analysis scenarios.

**Strengths:**

1.Innovative Approach: The paper creatively relates community detection to crystallization kinetics, introducing an annealing mechanism to enhance community core consistency and scalability with the novel CLANN model.

2.Scalability Improvement: The unsupervised Transitive Annealer significantly reduces computational complexity compared to GANs and reinforcement learning, improving performance on large-scale datasets.

3.Significant Performance Gains: Experiments show that CLANN outperforms existing methods in accuracy and efficiency across various real-world datasets, highlighting its broad applicability.

**Weaknesses:**

1.Questionable Validity of Physical Analogy: The analogy between community detection and crystallization kinetics is debatable, particularly regarding the relevance of energy minimization in community detection.

2.Insufficient Explanation of Crystallization Dependency: The paper does not adequately clarify how the introduced crystallization principles apply to community detection and their role in model optimization.

3.Limited Adaptability to Different Network Structures: CLANN’s effectiveness on non-clique structures or noisy communities is not thoroughly assessed, raising doubts about its generalizability.

4.Lack of Theoretical Analysis: The method lacks in-depth theoretical validation, such as proofs of convergence and complexity for the Transitive Annealer, along with insufficient sensitivity analyses for hyperparameters.

**Questions:**

1.Applicability of Physical Analogy: The analogy between community detection and crystallization lacks theoretical validation. Please clarify how crystallization principles (e.g., "stability," "cohesion") apply and impact core consistency and scalability.

2.Theoretical Support for CLANN: The model lacks theoretical analysis on convergence and complexity. Please provide complexity proof or detailed time complexity analysis, particularly for large datasets.

3.Parameter Sensitivity Analysis: The paper lacks a systematic sensitivity analysis of key hyperparameters (e.g., annealing temperature, loss weights). Please describe how performance varies with these parameters.

4.Applicability to Non-Clique Structures: CLANN’s reliance on cliques as cores raises questions about its adaptability to non-clique or noisy networks. Please test on such datasets to verify its general applicability.

---

> ### Author Response · Authors · 2024-11-23
>
> We appreciate the your insightful comment regarding the validity of physical analogy, adaptability, and sensitivity analysis. Please see our detailed one-by-one responses below.
>
> > ## W1+Q1. Questionable Validity of Physical Analogy
>
>  Energy minimization principles from physics have been effectively applied to network community detection. In previous reference in  **Global Response1**, maximizing modularity corresponds to minimizing system energy for stable community partitioning. Our study extends this link by showing that optimizing energy-based loss functions aligns with modularity optimization, resulting in reasonable community configurations (Page 14-15, Section A). Please refer to our reply in **Global Response 1** for more detailed explanation.
>
> > ## W2. Insufficient Explanation of Crystallization Dependency
>
>  Inspired by the analogy between crystallization and community detection, we leverage crystallization principles in CLANN to address two key challenges: community core inconsistency and growth scalability. Specifically, among the four crystallization principles:
> 1. The stability principle ensures that cores remain compact and representative;
> 2. The cohesion principle promotes strong internal connections within communities;
> 3. The growth analogy helps overcome barriers to meaningful expansion;
> 4. The equilibrium principle prevents overgrowth or fragmentation.
>
> These principles are seamlessly integrated into CLANN's loss functions, such as an energy-based loss to encourage cohesive embeddings and an interface-based loss to regulate expansion. This approach achieves enhanced consistency and scalability in community detection without relying on complex mechanisms like GANs or reinforcement learning. Moreover, we provide a crystallization video in an anonymous GitHub repository in **Global Response 1**.
>
> > ## W3+Q4. Limited Adaptability to Different Network Structures and Non-Clique Structures
>
>  We employed extra network datasets to demonstrate our CLANN's effectiveness. Concretely, we conducted experiments on **43 (7(Section 6 - Setting 1) + 15(Section 6 - Setting 2) + 6 (6 Bipartite Networks without Cliques) + 15 (Non-clique, Sparse, Noisy Networks)) different networks**. Here we also provide the results of CLANN's performance on newly introduced **Non-Clique, Sparse, and Noisy** networks as shown below. The detailed results and analysis can be found in the common response **Global Response 2**.
>
> ```
> Table 1. CLANN's F1 Performance Across 3 New Scenarios
> ```
> |Dataset|S 1K|S 2K|S 3K|S 4K|S 5K|B 1K|B 2K|B 3K|B 4K|B 5K|N 1K|N 2K|N 3K|N 4K|N 5K|
> |-|-|-|-|-|-|-|-|-|-|-|-|-|-|-|-|
> |#Node|64,020|100,000|100,000|100,000|100,000|100,000|100,000|100,000|100,000|100,000|25,968|47,318|70,425|97,270|215,912|
> |#Edge|66,665|200,000|200,000|200,000|200,000|200,000|200,000|200,000|200,000|200,000|88,949|181,547|280,586|397,201|829,877|
> |CLARE|0.1929|0.1522|0.1904|0.2153|0.2394|0.1291|0.1871|0.216|0.2458|0.2589|0.3308|0.3608|0.366|0.3753|0.2837|
> |CLANN|0.31|0.1567|0.2824|0.298|0.3011|0.1922|0.2513|0.2992|0.33|0.3307|0.4157|0.4475|0.4742|0.4398|0.3539|
>
>
> > ## W4+Q2+Q3. Lack of Theoretical Analysis and insufficient sensitivity analyses for hyperparameters, detailed time complexity analysis
>
>  Thanks for your valuable feedback. We agree that convergence proofs and complexity analyses for the Transitive Annealer, as well as sensitivity analyses for hyperparameters, are critical for a comprehensive understanding of the method. Here we provide brief explanation, for more detailed responses, please refer to our reply in **Global Response 3 and 4** .
>
> - **For Complexity Analysis**: We have included a detailed complexity analysis in **Appendix C**, showing that the time complexity of the entire framework is $O(N^2)$. The analysis breaks down the computational steps of the **Preliminary Core Filter**, **Nucleus Proposer**, and **Transitive Annealer**, demonstrating their efficiency in large-scale settings.
> - **For Convergence**: Regarding convergence, the **Transitive Annealer** is designed to terminate in at most $M$ steps, where $M$ is the maximum number of iterations. Empirical results in **Section 6.4** confirm that the annealer converges within 3 iterations in the vast majority of cases. This is supported by the case visualizations in **Figure 6**.
> - **For Hyperparameter Analysis**: In Section 6.5, we have conducted a sensitivity analysis of the 3 loss weights for the energy-based, interface-based, and consistency-based losses. In Appendix I, we studied the impact of the candidate size rate to analyze the quality of candidates selected by Nucleus Proposer.

---

> ### Comment · Area_Chair_fmVn · 2024-11-25
>
> Could please acknowledge and respond to the rebuttal.

---

> ### Author Response · Authors · 2024-11-27
>
> Dear Reviewer hnzb:
>
> We would like to express our sincere gratitude for your time in reviewing our paper and your valuable comments.
>
> Since the revision submission period is nearing its end, we are wondering whether our response has sufficiently addressed your questions. If you have any additional suggestions, we are more than willing to engage in further discussions and make necessary improvements to the paper.
>
> Thank you once again for dedicating your time to enhancing our paper!
>
> All the best,
>
> Authors

---

> ### Author Response · Authors · 2024-12-01
>
> Dear Reviewer hnzb,
>
> Since the End of author/reviewer discussions is coming in ONE day, may we know if our response addresses your main concerns? If so, we kindly ask for your reconsideration of the score. Should you have any further advice on the paper and/or our rebuttal, please let us know and we will be more than happy to engage in more discussion and paper improvements.
>
> Thank you so much for devoting time to improving our paper!

---

### Official Review · Reviewer_4nXW · 2024-11-03

**Soundness:** 3
**Presentation:** 3
**Contribution:** 3
**Rating:** 6
**Confidence:** 3

**Summary:**

The manuscript presents a novel method for community detection called CLANN. This method addresses challenges in semi-supervised community detection, particularly issues with community core consistency and scalability. The authors draw an analogy between community formation and crystallization kinetics, introducing two main components: the Nucleus Proposer and the Transitive Annealer. These components use cliques as starting points and leverage physics-based principles to optimize and expand communities effectively. The paper claims that CLANN surpasses existing state-of-the-art methods in accuracy and efficiency based on extensive experiments across various datasets.

**Strengths:**

1:The analogy with crystallization kinetics introduces a novel physics-grounded perspective for community detection, enriching the field with fresh conceptual insights.

2:Empirical evaluations show that CLANN outperforms established methods in both single and hybrid dataset scenarios, demonstrating its robustness and superior scalability.

3:The integration of the Nucleus Proposer and Transitive Annealer simplifies the growth process without relying on computationally heavy methods like GANs or reinforcement learning.

4:The paper provides extensive tests, including ablation studies and adaptability analyses, solidifying the validity of its contributions.

**Weaknesses:**

1:While the physics-based analogy is intriguing, it may be difficult for readers without a background in crystallization kinetics to grasp fully. More simplified explanations or visual aids could make this clearer.

2:Although the method improves scalability over GAN-based approaches, the process of clique enumeration can still be computationally expensive, potentially limiting applicability to extremely large graphs.

3:The experiments, though diverse, may benefit from including more real-world networks with varying characteristics to generalize the method's effectiveness further.

4:The paper does not thoroughly discuss the practical implementation details, such as computational resources required for different dataset sizes, which may affect the adoption of the model in real-world applications.

**Questions:**

1:Can you provide more intuitive examples or visual explanations of how crystallization kinetics translate to the community detection process?

2:How does CLANN compare with simpler heuristic-based community detection methods in terms of performance and resource consumption?

---

> ### Author Response · Authors · 2024-11-23
>
> Thank you very much for the constructive comments and questions. We are grateful for the positive comments on the novelty of design and empirical justification. Please see our detailed one-by-one responses below.
>
> > ## W1+Q1. More simplified explanations or visual aids of analogy
>
> To help reader to understand crystallization process better, we upload a visualization video to an anonymous GitHub repository  in **Global Response (Part 1)**. This video shows taht the crystallization process starts from some specific points (community cores in our case). By growing and merging (merging cliques in our case), the sub-grains grow into bigger crystalline structures (final communities in our case).
>
> > ## W2. Expensive computational of clique enumeration and limited applicability to extremely large graphs.
>
> Thanks for the constructive comments.  To validate the our model's efficiency, we conducted detailed computational complexity analysis in **Appendix Section C**, and runtime analysis in **Section 6.4 and Appendix Section J**. Also, please refer to our reply in **Global Response 4** about Computational Efficiency for more detailed response.
>
>
> > ## W3. More real-world networks with varying characteristics may benefit the generalization.
>
> Thank you very much for the constructive comments.  To validate the effectiveness, we conducted extra network datasets to demonstrate our CLANN's effectiveness. Concretely, we conducted experiments on **43 (7(Section 6 - Setting 1) + 15(Section 6 - Setting 2) + 6 (6 Bipartite Networks without Cliques) + 15 (Non-clique, Sparse, Noisy Networks)) different networks**. The detailed results and analysis can be found in the **Global Response 2**.
>
>
> > ## W4. Lack of some implementation details.
>
> Thanks for pointing out this problem. Most of the implementation detail are provided in Appendix H**. As noted, all experiments were conducted on an **AMD EPYC 7763 64-core Processor with 256GB of memory** and a **single NVIDIA RTX A5000 GPU (24GB memory)**. The overall size of the model (104,783 parameters, 424KB checkpoint size) and the average runtime for datasets of varying scales (reported in **Table 6** and **Figure 10**) indicate the model’s computational efficiency.
>
> > ## Q2. How does CLANN compare with simpler heuristic-based community detection methods in terms of performance and resource consumption?
>
> In the original submission, we provide the performance comparison with traditional unsupervised methods in Table 13 (Setting 1), and 15 (Setting 2). In the revised version, we also provide the runtime comparison in Table 14 (colored by blue). We put the performance and runtime comparison under Setting 1 here.
>
> ```
> Table 1: F1 Score Compared with Unsupervised Methods under Setting 1
> ```
> |Dataset|Est. α|SBM|N-SBM|O-SBM|Louvain|Label Prop.|Spectral|CLANN|
> |-|-|-|-|-|-|-|-|-|
> |A|12.91|0.3058|0.0371|0.0319|0.8226|0.7789|0.8226|**0.9055**|
> |D|2.86|0.0924|0.0000|0.0068|0.1986|0.3777|0.3070|**0.4701**|
> |L|3.21|0.1601|0.0000|N/A|0.4201|0.4801|0.4806|**0.5144**|
> |A/D|2.95|0.0847|0.0371|0.0055|0.1206|0.4710|0.1575|**0.6578**|
> |D/A|2.95|0.0876|0.0000|0.0057|0.1000|0.3606|0.1400|**0.4355**|
> |D/L|3.30|0.0332|0.0000|N/A|0.0599|0.3346|0.0946|**0.3373**|
> |L/D|3.30|0.1317|0.0000|N/A|0.2224|0.3849|0.3699|**0.3932**|
>
> - For Spectral clustering methods, the best results among DBSCAN, HDBSCAN, and OPTICS are presented.
> - Est. α: Estimated α value of the degree power-law fit.
> - N/A: Did not converge within 2 days.
>
> ```
> Table 2: Runtime (in Seconds) Comparison with Unsupervised Methods under Setting 1
> ```
> |Dataset|SBM|N-SBM|Louvain|Label Prop.|DBSCAN|HDBSCAN|OPTICS|CLANN|
> |-|-|-|-|-|-|-|-|-|
> |A|4.4|4.5|0.3|0.2|0.1|0.2|5.1|112.1|
> |D|23.3|109.1|3.2|1.9|1.5|3.0|51.4|291.3|
> |L|190.2|605.7|15.8|7.0|6.0|4.9|147.7|945.5|
> |A/D|30.2|132.5|4.6|2.2|2.1|3.7|70.3|717.0|
> |D/A|34.2|190.4|3.8|2.4|2.0|3.6|70.2|124.2|
> |D/L|224.9|856.9|20.9|11.0|19.2|8.8|314.5|453.5|
> |L/D|261.3|863.3|17.4|9.5|19.1|12.6|318.6|839.2|
>
> CLANN consistently outperforms heuristic-based methods like Louvain and Label Propagation, achieving the highest F1 scores across diverse datasets. For instance, on the Amazon dataset, CLANN scores 0.9055, significantly higher than Louvain's 0.8226. Its advantage is even more pronounced on hybrid datasets like "A/D," where it achieves 0.6578 compared to Louvain's 0.1206, showcasing its ability to handle complex community structures with superior accuracy and robustness.
>
> Although CLANN requires more computation time, its consistant better performance justifies the trade-off. On the Amazon dataset, it achieves far better results in 112.1 seconds compared to Louvain's 0.3 seconds. Features like the Nucleus Proposer and Transitive Annealer enable efficient refinement of community structures, making CLANN an ideal choice when accuracy outweighs runtime considerations.

---

> ### Comment · Area_Chair_fmVn · 2024-11-25
>
> Could please acknowledge and respond to the rebuttal.

---

> ### Author Response · Authors · 2024-11-27
>
> Dear Reviewer 4nXW:
>
> We would like to express our sincere gratitude for your time in reviewing our paper and your valuable comments.
>
> Since the revision submission period is nearing its end, we are wondering whether our response has sufficiently addressed your questions. If you have any additional suggestions, we are more than willing to engage in further discussions and make necessary improvements to the paper.
>
> Thank you once again for dedicating your time to enhancing our paper!
>
> All the best,
>
> Authors

---

> ### Author Response · Authors · 2024-12-01
>
> Dear Reviewer 4nXW,
>
> Since the End of author/reviewer discussions is coming in ONE day, may we know if our response addresses your main concerns? If so, we kindly ask for your reconsideration of the score. Should you have any further advice on the paper and/or our rebuttal, please let us know and we will be more than happy to engage in more discussion and paper improvements.
>
> Thank you so much for devoting time to improving our paper!

---

### Author Response · Authors · 2024-11-23
**Global Response (Part 1)**

Dear all reviewers, thank you very much for the time and effort in reviewing our paper, and for the constructive and positive comments. Our rebuttal consists of two parts: **Global Response** where we address the shared concerns from two or more reviewers and **Individual Response** where we provide a detailed one-to-one response to address your questions/concerns individually.

---
> ## **1. Analogy Rationale Between Crystallization Kinetics and Community Detection**

Applying energy minimization principles from physics to network community detection has proven to be valid and effective [Ref1, Ref2, Ref3]. Specifically, in [Ref3], the modularity function is shown to be equivalent to the Hamiltonian (energy function) of a Potts spin glass model, establishing a direct connection between modularity optimization and energy minimization. Maximizing modularity corresponds to finding a configuration of communities that minimizes the system's energy, leading to a more stable and natural partitioning of the network. In our study, we link our energy-based loss functions to modularity, demonstrating that optimizing these loss functions aligns with modularity optimization to achieve a reasonable community configuration (Page 14-15, Section A).

As outlined in the introduction (Line 50 - 72), core consistency refers to the embedding similarity between a community core and the corresponding complete community. Stability and cohesion principles, reflected in the Energy-Based Loss and Consistency-Based Loss, incorporate diverse information during community formation—such as size, misplacement, and shape—into the embeddings. This ensures that the community core embedding aligns closely with the complete community embedding. In contrast, previous methods neglect the community formation process, focusing solely on nodes or k-hop structures, which leads to inconsistent core embeddings.

For scalability, the principles of spontaneous growth and equilibrium status embed the natural community formation process directly into the model. This enables the community core to grow autonomously, eliminating the need for a second-stage, computationally expensive training process involving RL or GANs. These principles collectively enhance both the consistency and scalability of the model while maintaining efficiency and accuracy in community detection.

To help reader to understand crystallization process better, we upload our source code and a visualization video to an anonymous GitHub repository at https://anonymous.4open.science/r/CLANN_ICLR2025-51D0/README.md

**References**:
[Ref1] Pang, Y., & Li, K. (2013). An Energy Model for Network Community Structure Detection. *In Advanced Data Mining and Applications: 9th International Conference*.
[Ref2] Pang, Y., Bai, L., & Bu, K. (2015). An energy model for detecting community in PPI networks. *In International Conference on Data Management in Cloud, Grid and P2P Systems*.
[Ref3] Reichardt, J., & Bornholdt, S. (2006). Statistical mechanics of community detection. *Physical Review E—Statistical, Nonlinear, and Soft Matter Physics*.

---
> ## **2. Experiments on More Datasets**

In our original submission, we evaluated CLANN's performance on **non-clique** structures as shown in Appendix Section K. By replacing cliques with scatter-based cores, we tested the model on **6 bipartite datasets** (collaboration and co-purchase networks). These results demonstrate our robust performance under non-clique conditions. Additionally, experiments on **15 datasets** across three scenarios (**Non-Clique**, **Sparse**, and **Noisy**) further validate CLANN's effectiveness.
- **scatter datasets**, communities were generated with tree-structures to ensure **non-clique** structures.
- **Barabási-Albert datasets** are scale-free sparse networks with **fewer than 300 cliques**, simulating natural community growth.
- **NoisyDBLP datasets** introduced 10% edge perturbation for DBLP datasets while preserving community connectivity.

Altogether, CLANN was tested on **43 networks**, demonstrating its robustness in diverse and challenging conditions. Detalied results and analysis are presented in Appendix Section K (colored by blue).

```
Table 1. CLANN's F1 Performance Across 3 New Scenarios
```
|Dataset|S 1K|S 2K|S 3K|S 4K|S 5K|B 1K|B 2K|B 3K|B 4K|B 5K|N 1K|N 2K|N 3K|N 4K|N 5K|
|-|-|-|-|-|-|-|-|-|-|-|-|-|-|-|-|
|#Node|64,020|100,000|100,000|100,000|100,000|100,000|100,000|100,000|100,000|100,000|25,968|47,318|70,425|97,270|215,912|
|#Edge|66,665|200,000|200,000|200,000|200,000|200,000|200,000|200,000|200,000|200,000|88,949|181,547|280,586|397,201|829,877|
|CLARE|0.1929|0.1522|0.1904|0.2153|0.2394|0.1291|0.1871|0.216|0.2458|0.2589|0.3308|0.3608|0.366|0.3753|0.2837|
|CLANN|0.31|0.1567|0.2824|0.298|0.3011|0.1922|0.2513|0.2992|0.33|0.3307|0.4157|0.4475|0.4742|0.4398|0.3539|

These results demonstrate that CLANN adapts effectively to non-clique, sparse, and noisy structures while maintaining superior performance.

---

### Author Response · Authors · 2024-11-23
**Global Response (Part 2)**

> ## **3. Sensitivity Analyses of Hyperparameters**
1. Existing Sensitivity Analysis:
In **Section 6.5**, we have conducted a sensitivity analysis of the **3 loss weights** for the energy-based, interface-based, and consistency-based losses. As shown in **Figure 5**, we normalized these losses to the same magnitude and varied their weights across a wide range of values (0.01, 0.1, 1, 10, 100). The results demonstrate that the model’s performance remains stable, highlighting its robustness to changes in loss weights. In **Appendix I**, we studied the impact of the **candidate size rate** to analyze the quality of candidates selected by Nucleus Proposer.

2. For annealing temperature and transtive threshold in Algorithm 2: In the real deployment, we try to avoid overusing many hyperparameters to improve model's generalizability,
- **Annealing Temperature Threshold**: In the deployment, it is set to be a temperature probability function as $P_{\text{temp}}(|S_i^c|) = \Phi\left( \frac{|S_i^c| - \mu}{\sigma} \right)$. It represents a cumulative distribution function of a normal distribution where $\mu$ is the mean and $\sigma$ is the standard deviation of the training community size. In this case, this threshold can dynamically adapt to different datasets.
- **Transitive threshold**: As explained in Section 5.3, we just simply check whether the integrity score of the newly merged clique is higher than that of the previous state, making the transitive threshold unnecessary.

Consider the generalizability and to avoid confusion, in the revised version, we have removed both parameters from Algorithm 2, and added illustration before euqation (13).


---
> ## **4. Computational Efficiency**
1. **Scalability of Clique Enumeration**:
As detailed in **Appendix C and Appendix F**, we avoid traversing the entire graph to enumerate all cliques $\(O(3^{N/3})\)$, instead, we extract maximum cliques from 1-hop ego networks for the pre-selected node. We only check the clique for this node among its 1-hop ego network, reducing complexity to $\(O(N^2)\)$. In the real deployment, after optimizing our source code, we achieved faster speed and less RAM requirement compared to the runtime reported in our original version. The source code and datasets are provided in the anonymous GitHub repository mentioned in the previous Global Response.

2. **Theoretical Validation**:
For complexity and convergence, we provide a detailed complexity analysis in **Appendix C**, breaking down the computational steps for the **Preliminary Core Filter**, **Nucleus Proposer**, and **Transitive Annealer**. The Transitive Annealer is guaranteed to terminate within \(M\) iterations, as confirmed by empirical results in **Section 6.4**, where most cases converge in fewer than 3 iterations (**Figure 6**).

3. **Efficiency of the Candidate Filtering Mechanism**:
- **Efficiency of the Preliminary Core Filter**: The Preliminary Core Filter pre-selects candidate community cores by only leveraging a lightweight classifier (input vector dimension is only **5**). The standard augment feature for this filter only involves simple operations (**degree calculation**), which are computationally lightweight and can be performed in near real-time. This approach ensures the filtering mechanism remains efficient and scalable, even in real-time or streaming scenarios.

- **Real-Time Adaptation for New Nodes**: For dynamic graphs where new nodes are added, the computational requirement is further minimized. When a new node comes in, instead of recomputing all existing nodes, we only update very basic standard augment feature for the related neighbor nodes (degree-based features, Page 6 Line 272-273), because the new node only connects to a few existing nodes, which will not change the standard augment feature of most unconnected nodes.

---

### Meta-Review · Area_Chair_fmVn · 2024-12-19

**Metareview:**

Overall, the paper seems difficult for the machine learning community to understand. The reviewers reported multiple issues, which unfortunately the rebuttal did not address. I cannot accept the paper at this stage. The authors should put more effort into communicating with the machine learning community by modifying the paper for this audience.

**Additional Comments On Reviewer Discussion:**

See my comments above.

---

### Decision · Program_Chairs · 2025-01-22

Reject